# Where to Pay Attention in Sparse Training for Feature Selection?

**Ghada Sokar**
Eindhoven University of Technology
`g.a.z.n.sokar@tue.nl`

**Zahra Atashgahi**
University of Twente
`z.atashgahi@utwente.nl`

**Mykola Pechenizkiy**
Eindhoven University of Technology
`m.pechenizkiy@tue.nl`

**Decebal Constantin Mocanu**
University of Twente
Eindhoven University of Technology
`d.c.mocanu@utwente.nl`

## Abstract

A new line of research for feature selection based on neural networks has recently emerged. Despite its superiority to classical methods, it requires many training iterations to converge and detect informative features. The computational time becomes prohibitively long for datasets with a large number of samples or a very high dimensional feature space. In this paper, we present a new efficient unsupervised method for feature selection based on sparse autoencoders. In particular, we propose a new sparse training algorithm that optimizes a model's sparse topology during training to pay attention to informative features quickly. The attention-based adaptation of the sparse topology enables fast detection of informative features after a few training iterations. We performed extensive experiments on 10 datasets of different types, including image, speech, text, artificial, and biological. They cover a wide range of characteristics, such as low and high-dimensional feature spaces, and few and large training samples. Our proposed approach outperforms the state-of-the-art methods in terms of selecting informative features while reducing training iterations and computational costs substantially. Moreover, the experiments show the robustness of our method in extremely noisy environments[1].

## 1 Introduction

Feature selection plays a crucial role in data mining and machine learning tasks with the explosion in the size and dimensionality of real-world data [68; 10; 30; 44; 23]. It aims to efficiently select a subset of the features that are most informative and remove the irrelevant or redundant ones. It is useful for alleviating the curse of dimensionality, interpretability of model-driven decisions, generalization of downstream tasks, reducing the prohibitive memory and computational costs, and avoiding the expensive costs of collecting the full set of features as in biological studies [11; 38; 31; 52].

Many methods have been proposed for feature selection in supervised, semi-supervised, and unsupervised settings [75; 10; 60; 51; 19; 1]. Recently, using neural networks for feature selection has received increasing attention due to its power to learn non-linear dependencies among input features [39]. One limitation of current neural network-based methods is that they are computationally expensive. The computational costs result from training *dense* models for *many* training iterations until informative features are recognized. The costs increase for datasets with a very large number of training samples or high dimensional feature space. This limitation is recently addressed in [4] by utilizing sparse networks with dynamic sparsity for feature selection. An autoencoder is trained with

---

[1]Code is available at https://github.com/GhadaSokar/WAST.

the sparse training algorithm SET [54], which explores different sparse topologies during training by dropping a portion of the connections and growing the same portion *randomly*. The importance of an input feature is estimated after convergence based on its outgoing sparse connections. Training sparse networks from scratch decreases memory and computational costs compared to previous dense-based methods. Yet, the *random* exploration of sparse topologies requires many training iterations to identify informative features.

In this paper, we propose an efficient unsupervised method for feature selection. We introduce a new sparse training algorithm for autoencoders that quickly detects the informative input features. Specifically, we optimize the sparse topology to learn **W**here to pay **A**ttention during **S**parse **T**raining. We named our method **WAST**. WAST exploits the information from the reconstruction loss and the weights of sparse connections to guide the topological search. It redistributes the sparse connections to spot informative features quickly. We evaluate our proposed approach on ten datasets from various domains, including image, speech, text, artificial, and biological. Through extensive experiments, we find that WAST identifies the informative features after a few training iterations and outperforms state-of-the-art unsupervised feature selection methods. Our main contributions are:

- We propose a new efficient unsupervised method for feature selection, named WAST. WAST optimizes the sparse topology of an autoencoder to detect the informative features quickly.
- We perform extensive experiments on 10 benchmarks that cover various types and characteristics. Experimental results show the effectiveness of our method over the state-of-the-art methods in terms of selecting the informative features.
- WAST reduces the computational time compared to neural network-based methods substantially by reducing training iterations and employing highly sparse neural networks.
- We illustrate the robustness of our method in extremely noisy environments and its effectiveness for datasets with very high dimensional feature space and a few training samples.

## 2 Related Work

### 2.1 Feature Selection

Many feature selection methods were introduced for the supervised, semi-supervised, and unsupervised settings depending on whether the data is labeled, partially labeled, or unlabeled, respectively [75; 10; 60; 1]. Typically, feature selection methods are divided into three categories: filter, wrapper, and embedded methods. Filter methods are independent of the learning algorithms. They use ranking techniques for providing scores to the features, such as Laplacian score [26]. Despite being fast, they do not consider the relationship between the features, which may result in the selection of redundant features [11]. In the supervised setting, filter methods consider the relation between the feature and the class label, such as fisher_score [22], CIEF [41], and ICAP [33]. Wrapper methods exploit the performance of a predictive model to evaluate the quality of a subset of the features [70; 12; 74]. They are more effective than filter methods, yet they are computationally expensive [39]. Embedded methods incorporate feature selection into the learning phase of another algorithm. Multi-Cluster Feature Selection (MCFS) [9] uses regularization to select the best features that keep the multi-cluster structure of the data. Unsupervised Discriminative Feature Selection (UDFS) [72] uses $l_{2,1}$ regularization and discriminative analysis to select the most discriminative features. Similarly, in the supervised setting, RFS [56] and L1_L21 [45] exploit $l_{2,1}$-norm to introduce feature sparsity.

Another direction under the embedded category has recently emerged. It uses neural networks to perform feature selection by learning the non-linear dependencies among input features [40]. The success of autoencoders as a tool for feature extraction encourages unsupervised methods to explore their power for feature selection [7; 6; 67; 25; 18]. AutoEncoder Feature Selector (AEFS) [25] combines autoencoder regression and group lasso tasks. Concrete Autoencoder (CAE) [5] learns a concrete selector layer (encoder) that selects stochastic linear combinations of input features during training. After training, it converges to the target number of features. Despite the high performance of these methods, the large number of iterations required to train dense models increases the computational costs significantly. The conceptually closest work to ours is the recent work, Quick Selection (QS) [4]. It trains a sparse autoencoder from scratch. During training, different sparse topologies are explored using the SET method [54]. After each *training epoch* (i.e., pass on the full data), a fraction of the connections with the least magnitude is dropped, and the same fraction

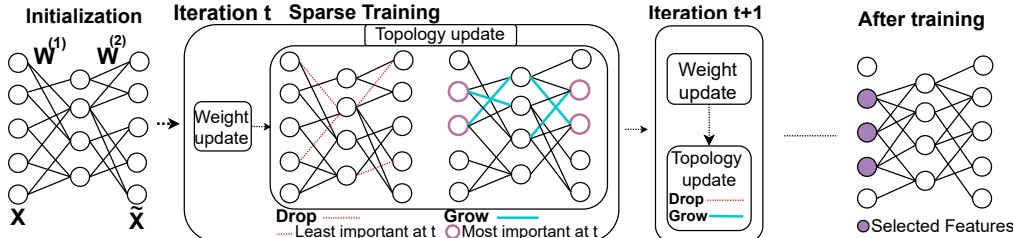

Figure 1: An overview of our proposed method WAST for unsupervised feature selection. A sparse autoencoder is initialized with uniformly distributed sparse connections. During sparse training, the connections are redistributed in the most important neurons at iteration $t$ during the "drop-and-grow" cycle. After convergence, neurons with the highest importance are selected as the most informative features.

is randomly regrown. Few works based on multi-layer perceptron networks are proposed for the supervised setting [37; 61; 71]. In section 4, we denote non Neural Network-based (NN-based) methods as classical methods.

## 2.2 Sparse Training

Sparse training with dynamic sparsity, also known as Dynamic Sparse Training (DST), is a recent research direction that aims at accelerating the training of neural networks without sacrificing performance [27; 54; 47]. A neural network is initialized with a random sparse topology from *scratch*. The sparse topology (connectivity) and the weights are jointly optimized during training. During training, the sparse topology is changed periodically through a *drop-and-grow* cycle, where a fraction of the parameters are dropped, and the same portion is regrown among different neurons. An update schedule determines the frequency of topology updates. Many works have been proposed, focusing on improving the performance of sparse training for supervised image classification tasks by introducing different criteria for connection regrowth [20; 34; 59; 49; 73; 50]. DST demonstrated its success in other fields as well, such as continual learning [62], feature selection [4], ensembling [46], federated learning [77; 8], adversarial training [57], and deep reinforcement learning [63].

## 3  Where to pay Attention in Sparse Training (WAST)?

**Problem Formulation.** Let $\mathcal{D}$ be a dataset with samples $\{\mathbf{x}^{(j)}\}_1^n$, where $\mathbf{x}^{(j)} \in \mathbb{R}^m$, and $m$ and $n$ are the number of features and samples, respectively. The goal is to select a subset of the features of a size $K$ in an *unsupervised* manner, which is most representative of the whole feature space $m$.

WAST is a new sparse training method for neural network-based unsupervised feature selection. The basic idea of WAST is to pay attention to input features based on their estimated importance during training to detect the most informative features quickly. An overview of WAST is shown in Figure 1. We use a *sparse* autoencoder with a *single* hidden layer of $h$ neurons. Let $f_{\mathbf{W}}$ be an autoencoder model parameterized by sparse weights $\mathbf{W} = \{\mathbf{W}^{(1)}, \mathbf{W}^{(2)}\}$. The network is initialized with a certain sparsity level $s = 1 - \frac{\|\mathbf{W}^{(1)}\|_0}{m \times h}$, where $\|.\|_0$ is the $L_0$ norm. The sparsity level is kept fixed during training. Initially, the sparse weights are uniformly distributed on the neurons. During training, besides weights optimization, the sparse topology is optimized such that the sparse connections are gradually redistributed on the most informative features. To achieve this goal, we propose new criteria to update the sparse topology and a new update schedule, as we will explain next.

**Training.** An autoencoder model is trained to minimize the reconstruction loss. We use mean squared error to measure the loss between an input sample $\mathbf{x}$ and the reconstructed one $\tilde{\mathbf{x}}$ as follows:

$$L = \|\tilde{\mathbf{x}} - \mathbf{x}\|_2^2, \tag{1}$$

where $\tilde{\mathbf{x}} = f_{\mathbf{W}}(\mathbf{x}) = \sigma(\mathbf{x}\mathbf{W}^{(1)})\mathbf{W}^{(2)}$, and $\sigma$ is a non-linear activation function. After each weight update using a batch of the data, we adapt the sparse topology. The effect of the update schedule on performance is shown in Appendix E. The sparse topology is updated through a **drop-and-grow** cycle.

---
**Algorithm 1** WAST
---

1: **Input:** Autoencoder network $f_{\mathbf{W}}$ with one hidden layer of size $h$, Dataset $\mathcal{D}$ with $m$ features
2: **Input:** Learning rate $\eta$, target number of features $K$, sparsity level $s$, $\lambda$, $\alpha$
3: **Output:** Indices of selected features from the $m$ features ($\mathcal{R}$)
4: Initialize $\mathbf{W} = \{\mathbf{W}^{(1)}, \mathbf{W}^{(2)}\}$ with sparsity level $s$
5: $\mathcal{I}_{I_i} \leftarrow 0 \quad \mathcal{I}_{O_i} \leftarrow 0 \quad \forall i \in \{1, ..., m\}$
6: **for** each training step $t$ **do**
7: $\quad$ Sample a Batch $B_t \sim \mathcal{D}$ with size $b$
8: $\quad \tilde{\mathbf{x}}^{(j)} \leftarrow f_{\mathbf{W}}(\mathbf{x}^{(j)}), \quad \forall \mathbf{x}^{(j)} \in B_t$
9: $\quad L \leftarrow \frac{1}{b} \sum_{j=0}^{b} \left\| \tilde{\mathbf{x}}^{(j)} - \mathbf{x}^{(j)} \right\|_2^2$
10: $\quad \mathbf{W} \leftarrow \mathbf{W} - \eta \nabla_{\mathbf{W}} L$
11: $\quad \mathcal{I}_{O_i} \leftarrow \mathcal{I}_{O_i} + \lambda |\frac{\partial L}{\partial \tilde{\mathbf{x}}_i}| + (1 - \lambda) \sum_{k=1}^{h} |\mathbf{W}_{ki}^{(2)}| \quad \forall i \in \{1, ..., m\}$
12: $\quad \mathcal{I}_{I_i} \leftarrow \mathcal{I}_{I_i} + \lambda |\frac{\partial L}{\partial \tilde{\mathbf{x}}_i}| + (1 - \lambda) \sum_{k=1}^{h} |\mathbf{W}_{ik}^{(1)}| \quad \forall i \in \{1, ..., m\}$
13: $\quad \mathbf{W}^{(l)} \leftarrow \text{Drop} (|\mathbf{W}^{(l)}|, \mathcal{I}_O, \mathcal{I}_I, \alpha) \quad \forall l \in \{1, 2\}$
14: $\quad \mathbf{W}^{(l)} \leftarrow \text{Grow} (\mathcal{I}_O, \mathcal{I}_I, \alpha) \quad \forall l \in \{1, 2\}$
15: **end for**
16: $\mathcal{R} \leftarrow top(\mathcal{I}_I, K)$

---

A fraction $\alpha$ of the connections with the least importance is dropped from each layer $\{\mathbf{W}^{(1)}, \mathbf{W}^{(2)}\}$, and the same portion of new connections are regrown. New connections are initialized to zero.

The main contribution of this work lies in where the new connections are regrown. Different from the previous method (QS) [4], where the connections are randomly regrown, we optimize the topology to pay fast attention to informative features during training. To this end, we measure the importance of each neuron in the input and output layers by leveraging the information available during training. The importance of a neuron is estimated by its impact on the reconstructed loss and the magnitude of its connected weights.

Specifically, we measure how sensitive the reconstruction loss $L$ is to changes in the reconstructed output $\tilde{\mathbf{x}}$. The first-order approximation for the change in loss with respect to a small perturbation $\delta = \{\delta_i\}$ in the output $\tilde{\mathbf{x}} = \{\tilde{\mathbf{x}}_i\}$ can be written as a sum of its individual components as follows:

$$L(\tilde{\mathbf{x}} + \delta) - L(\tilde{\mathbf{x}}) \approx \sum_{i=1}^{m} \frac{\partial L}{\partial \tilde{\mathbf{x}}_i} \delta_i, \tag{2}$$

where $\frac{\partial L}{\partial \tilde{\mathbf{x}}_i}$ is the gradient of the loss with respect to the output neuron $\tilde{\mathbf{x}}_i$. We define the importance of a neuron $i$ in the output layer at training iteration $t$ as:

$$\mathcal{I}_{O_i}^{(t)} = \mathcal{I}_{O_i}^{(t-1)} + \lambda |\frac{\partial L}{\partial \tilde{\mathbf{x}}_i}| + (1 - \lambda) \sum_{j=1}^{h} |\mathbf{W}_{ji}^{(2)}|, \tag{3}$$

where $|\mathbf{W}_{ji}^{(2)}|$ is the magnitude of the incoming connection from neuron $j$ to neuron $i$ and $\lambda$ is a hyperparameter coefficient to balance the two components (See Appendix A). The same criterion is applied for the input neurons ($\mathcal{I}_{I_i}^{(t)}$), except that the magnitude of input weights ($|\mathbf{W}^{(1)}|$) is used instead of the output weights. This criterion considers two cases: the information from large gradients, especially at the beginning of the training, and the large weights resulting from the optimization at later stages. The new connections in the input and output layers are regrown on the neurons with the highest importance. For instance, the new connections in the input layer are given by $top_{i \notin \tilde{\mathbf{W}}^{(1)}}(\mathcal{I}_I \times \mathcal{I}_H^{\top}, r)$, where $top(Q, k)$ gives the indices of the top-$k$ elements in a matrix $Q$, $\mathcal{I}_H$ is the importance of the hidden neurons, $\tilde{\mathbf{W}}^{(1)}$ is the set of remaining weights after the drop phase, and $r$ is the number of regrown connections based on the fraction $\alpha$. We consider the neurons in the hidden layer to be equally important.

Table 1: The characteristics of datasets.

| Type | Dataset | #Features | #Classes | #Samples | #Train | #Test |
|------|---------|-----------|----------|----------|--------|-------|
| Artificial | Madelon [24] | 500 | 2 | 2600 | 2000 | 600 |
| Image | UPSP [32] | 256 | 10 | 9298 | 7438 | 1860 |
| | COIL-20 [55] | 1024 | 20 | 1440 | 1152 | 288 |
| | MNIST [36] | 784 | 10 | 70000 | 60000 | 10000 |
| | Fashion MNIST [69] | 784 | 10 | 70000 | 60000 | 10000 |
| Speech | Isolet [21] | 617 | 26 | 7737 | 6237 | 1560 |
| Time Series | HAR [2] | 516 | 6 | 10299 | 7352 | 2947 |
| Text | PCMAC [35] | 3289 | 2 | 1943 | 1544 | 389 |
| Biological | SMK-CAN-187 [64] | 19993 | 2 | 187 | 149 | 38 |
| | GLA-BRA-180 [65] | 49151 | 4 | 180 | 144 | 36 |

During the drop phase, the importance of a connection $w$ in the input (output) layer is estimated by its magnitude and the importance of its connected input (output) neuron ($\mathcal{I}_i^{(t)}$) as follows:

$$\mathcal{I}_w^{(t)} = |w|\mathcal{I}_i^{(t)}, \tag{4}$$

where $\mathcal{I}_i^{(t)}$ takes the value of $\mathcal{I}_{I_i}^{(t)}$ and $\mathcal{I}_{O_i}^{(t)}$ for the input and output layers, respectively. The effect of each component in the neuron and connection importance criteria is studied in Section 5.3. The periodic update of the sparse topology redistributes the connections on the effective features.

**Feature Selection.** After training, we select the top-$K$ neurons with the highest importance from the input layer as the most informative $K$ features. The details of WAST are provided in Algorithm 1.

# 4 Experiments

## 4.1 Baselines

We compared our method with several autoencoder-based (QS [4], CAE [5], AEFS [25]) and classical methods (lap_score [26], MCFS [9], DUFS [42]) for unsupervised feature selection. On top of that, we include comparisons with seven state-of-the-art supervised feature selection methods. Although the latter is not the focus of this paper and not a typical comparison in the literature, we prefer to show that WAST performs competitively also with the supervised methods while being computationally efficient and applicable for cases where labels are not available or expensive to collect.

## 4.2 Datasets

We evaluate our method on 10 publicly available datasets, including image, speech, text, time series, biological, and artificial data. They have a variety of characteristics, such as low and high-dimensional features and a small and large number of training samples. Details are in Table 1.

## 4.3 Experimental Settings

**Evaluation Metrics.** An efficient feature selection method should have high learning accuracy with less memory and computational costs [10]. Hence, we evaluate multiple metrics. **(1) Classification Accuracy**: It is typically evaluated by a machine learning model [10]. We trained a classifier using the $K$ selected features by the studied methods. Here, we use one of the most popular classifiers, support vector machines (SVM) [14]. Other classifiers could be used. Yet, we choose to use a non-NN-based classifier for evaluation to avoid any biased advantages towards the NN-based baselines over the others. We studied the performance of 6 different values of the number of selected features ($K$). We report the average accuracy of the test data over 5 seeds. **(2) Memory cost:** We calculate the number of network parameters (#params) used by each NN-based method. **(3) Computational cost:** We calculate the number of Floating-point operations (FLOPs) consumed to train a neural network. See Appendix A.2 for details.

**Implementation.** We implemented WAST and QS [4] with PyTorch [58] (see Appendix H). For CAE [5] and AEFS [25], we used the code provided by the authors of CAE with MIT license[2]. For DUFS

---
[2]https://github.com/mfbalin/Concrete-Autoencoders

Table 2: Classification accuracy (%) ($\uparrow$) using unsupervised and supervised feature selection methods. 50 features are selected for all datasets except Madelon, where 20 features are used. The best performer from the unsupervised methods is indicated in bold font, while the best performer from the supervised methods is in blue.

| | | Method | Madelon | USPS | COIL-20 | MNIST | FashionMNIST |
|---|---|---|---|---|---|---|---|
| Unsupervised | Classical | lap_score [26] | 49.50±0.00 | 70.54±0.00 | 78.12±0.00 | 23.94±0.00 | 27.07±0.00 |
| | | MCFS [9] | 51.83±0.00 | 93.33±0.00 | 97.22±0.00 | - | - |
| | | DUFS [42] | 52.57±1.35 | 95.62±0.54 | 97.43±1.22 | 62.09± 0.00 | 74.69±1.86 |
| | NN-based | AEFS [25] | 60.16±4.61 | 95.86±0.48 | 99.48±0.41 | 93.22±1.38 | 80.88±0.71 |
| | | CAE [5] | 80.90±2.86 | 95.04±0.59 | 94.54±2.92 | **96.20±0.14** | **84.66±0.16** |
| | | QS [4] | 82.07±1.10 | 95.88±0.31 | 99.17±0.42 | 94.07±0.04 | 82.65±0.38 |
| | | WAST (ours) | **83.27±0.63** | **96.69±0.27** | **99.58±0.14** | 95.27±0.26 | 82.16±0.57 |
| Supervised | Classical | Fisher_score [22] | 75.67±0.00 | 91.02±0.00 | 88.89±0.00 | 86.11±0.00 | 67.85±0.00 |
| | | L1_L21 [45] | 49.33±0.00 | 86.99±0.00 | 92.01±0.00 | 62.26±0.00 | 69.57±0.00 |
| | | CIFE [41] | 54.50±0.00 | 61.29±0.00 | 59.38±0.00 | 89.30±0.00 | 66.86±0.00 |
| | | ICAP [33] | 78.00±0.00 | 95.22±0.00 | 99.31±0.00 | 89.03±0.00 | 59.52±0.00 |
| | | RFS [56] | 83.00±0.00 | 95.32±0.00 | 66.32±0.00 | - | - |
| | NN-based | LassoNet [37] | 79.50±1.22 | 95.80±0.12 | 95.83±1.18 | 94.38±0.12 | 82.63±0.23 |
| | | STG [71] | 59.53±1.90 | 95.78±0.60 | 97.57±1.70 | 92.53±0.86 | 83.32±0.45 |
| | | Method | Isolet | HAR | PCMAC | SMK | GLA |
| Unsupervised | Classical | lap_score [26] | 75.71±0.00 | 82.80±0.00 | 49.87±0.00 | 81.58±0.00 | 66.67±0.00 |
| | | MCFS [9] | 81.41±0.00 | 80.29±0.00 | 53.47±0.00 | 78.95±0.00 | 75.00±0.00 |
| | | DUFS [42] | **85.62±2.53** | 86.90±1.06 | 57.79±3.18 | 81.05±3.07 | 70.83±1.39 |
| | NN-based | AEFS [25] | 80.94±2.02 | 89.54±0.44 | 60.40±2.37 | 79.48±3.07 | 67.76±6.21 |
| | | CAE [5] | 78.90±1.24 | 86.26±2.41 | 55.08±0.00 | 78.94±2.37 | 70.56±4.50 |
| | | QS [4] | 74.62±2.12 | 89.68±0.38 | 55.78±3.25 | 81.58±3.72 | 68.89±4.78 |
| | | WAST (ours) | 85.33±1.39 | **91.20±0.16** | **60.51±2.53** | **84.74±1.05** | **75.56±4.08** |
| Supervised | Classical | Fisher_score [22] | 75.64±0.00 | 83.68±0.00 | 86.38±0.00 | 73.68±0.00 | 63.89±0.00 |
| | | L1_L21 [45] | 55.90±0.00 | 81.30±0.00 | 53.98±0.00 | 84.21±0.00 | 69.44±0.00 |
| | | CIFE [41] | 59.81±0.00 | 84.15±0.00 | 75.84±0.00 | 81.58±0.00 | 58.33±0.00 |
| | | ICAP [33] | 75.06±0.00 | 88.70±0.00 | 87.66±0.00 | 73.68±0.00 | 72.22±0.00 |
| | | RFS [56] | 77.31±0.00 | 88.23±0.00 | 67.61±0.00 | 76.32±0.00 | - |
| | NN-based | LassoNet [37] | 85.70±0.38 | 93.93±0.15 | 86.53±1.25 | 77.37± 3.57 | 76.67±2.22 |
| | | STG [71] | 89.38±1.19 | 91.75±0.59 | 56.04±1.90 | 81.05±1.29 | 71.11±2.83 |

[42], LassoNet [37], and STG [71], we used the official codes with MIT license[3][4][5]. For classical baselines, we used the Scikit-Feature library with GNU General Public license [39][6]. NN-based and classical methods are trained on Nvidia GPUs and CPUs, respectively. We consider a 12 hours limit on the running time of each experiment. Experiments that exceed this limit are not considered.

**Implementation Details.** For all NN-based methods except CAE [5], we use a single hidden layer of 200 neurons. The architecture of CAE consists of two layers. The size of the hidden layers is dependent on the chosen $K$; $[K, \frac{3}{2}K]$. For WAST and QS, we use a sparsity level of 0.8. Following [4], we report the accuracy of NN-based baselines after 100 epochs unless stated otherwise. Note that some baselines reported a higher number of epochs (e.g., CAE [5] uses 200 epochs for some datasets). Yet, we keep 100 epochs for all cases for a fair comparison. For WAST, we train the model for 10 epochs. Following [4], we add a Gaussian noise with a factor of 0.2 to the input in WAST and QS [4]. Details of the hyperparameters are in Appendix A.1.

### 4.4 Results

**Accuracy.** Table 2 shows the classification accuracy of the studied datasets. Here, we report the challenging case where a few best informative features have to be selected. We use a $K$ of 50 for all datasets except for Madelon, where a $K$ of 20 is used as the remaining 480 features are pure noise. Experiments with various values for $K \in \{25, 50, 75, 100, 150, 200\}$ can be found in Appendix B, with a summary provided next.

Unsupervised NN-based methods outperform the classical ones on all datasets except one, where the performance difference is marginal. WAST outperforms unsupervised NN-based methods on

---

[3]https://github.com/Ofirlin/DUFS

[4]https://github.com/lasso-net/lassonet

[5]https://github.com/runopti/stg

[6]https://jundongl.github.io/scikit-feature/

Table 3: Memory and computational costs ($\downarrow$) estimated by the #params and FLOPs ($10^{12}$), respectively, for NN-based methods. Unlike other methods, the architecture of CAE [5] is dependent on the value of $K$. Here, we report the costs for $K = 50$.

| Method | Type | (Fashion)MNIST #params | (Fashion)MNIST FLOPs | USPS #params | USPS FLOPs | COIL-20 #params | COIL-20 FLOPs | Isolet #params | Isolet FLOPs |
|---|---|---|---|---|---|---|---|---|---|
| AEFS[25] | Dense | 313600 | 11.28 | 102400 | 0.45 | 409600 | 0.28 | 246800 | 0.92 |
| CAE[5] | Dense | 101750 | 3.66 | 35750 | 0.15 | 131750 | 0.09 | 80875 | 0.30 |
| QS[4] | Sparse | **62720** | 2.25 | **20480** | 0.09 | **81920** | 0.05 | **49360** | 0.18 |
| WAST | Sparse | **62720** | **0.22** | **20480** | **0.009** | **81920** | **0.005** | **49360** | **0.01** |

| Method | Type | HAR #params | HAR FLOPs | PCMAC #params | PCMAC FLOPs | SMK #params | SMK FLOPs | GLA #params | GLA FLOPs |
|---|---|---|---|---|---|---|---|---|---|
| AEFS[25] | Dense | 206400 | 0.91 | 1315600 | 1.22 | 7997200 | 0.71 | 19660400 | 1.69 |
| CAE[5] | Dense | 68250 | 0.30 | 414875 | 0.38 | 2502875 | 0.22 | 6147625 | 0.53 |
| QS[4] | Sparse | **41280** | 0.18 | **263120** | 0.24 | **1599440** | 0.14 | **3932080** | 0.33 |
| WAST | Sparse | **41280** | **0.01** | **263120** | **0.02** | **1599440** | **0.01** | **3932080** | **0.03** |

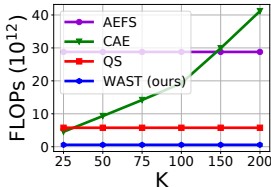

Figure 2: FLOPs ($\downarrow$) required for all datasets except Madelon at different values of $K$.

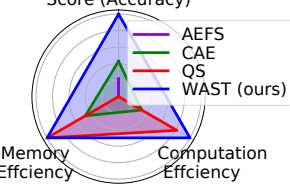

Figure 3: Comparison using various evaluation metrics computed on all datasets and 6 different values for $K$.

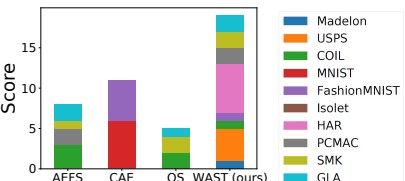

Figure 4: Score ($\uparrow$) (how many times a method is the best performer) of different methods on each dataset for 6 different values of $K$.

all datasets except MNIST and Fashion MNIST, where CAE is the best performer. It is worth emphasizing that the superior performance of WAST is achieved with a 90% reduction in the number of training iterations. Supervised baselines outperform the classical supervised ones in 6 cases. Interestingly, WAST achieves a competitive performance to the supervised baselines. It outperforms the best-supervised performer in 5 cases.

**Memory and Computational Costs.** Table 3 shows the memory and computational costs consumed by each unsupervised NN-based method. The high memory and computational costs are caused by the high number of training samples, as in MNIST, or the high-dimensional feature space, as in GLA. Training a sparse model from scratch, as in WAST and QS, reduces these costs substantially. QS reduces the computational cost by 80%. Interestingly, besides the reduction resulting from using sparse neural networks, WAST reduces the computational cost further to 98% due to the fast identification of the informative features. Both WAST and QS reduce the network size by 80%.

The network size of CAE is dependent on the target value of $K$. This has two limitations: training the model for every target $K$ and the increase of the memory and computational costs with higher values for $K$. On the other hand, WAST derives the importance of each feature with a single training run independently of the target $K$. This is illustrated in Figure 2. The figure shows the accumulated FLOPs for all datasets except Madelon at each value of the studied $K$.

**Overall Performance.** To give a holistic view of the performance of each method across all dimensions (accuracy, memory cost, and computational cost) on all datasets and all studied values of $K$, we calculate the performance in each case. Figure 3 illustrates the normalized performance using min-max scaling for all cases. The "score" metric represents the number of times a method is the best performer in terms of accuracy. Details are provided in Appendix A.2. CAE has a higher score than AEFS, while the former is the best only for image datasets with a large number of samples (i.e., MNIST and Fashion MNIST), as shown in Figure 4. AEFS has the highest memory and computational costs. QS improves the memory and computational costs but has a lower score. WAST has the highest score and spans more dataset *types* with different *characteristics* (e.g., high dimensional data) than the baselines (Figure 4). The performance gain is accompanied by a significant improvement in memory and computational efficiency. Detailed results are provided in Appendix B.

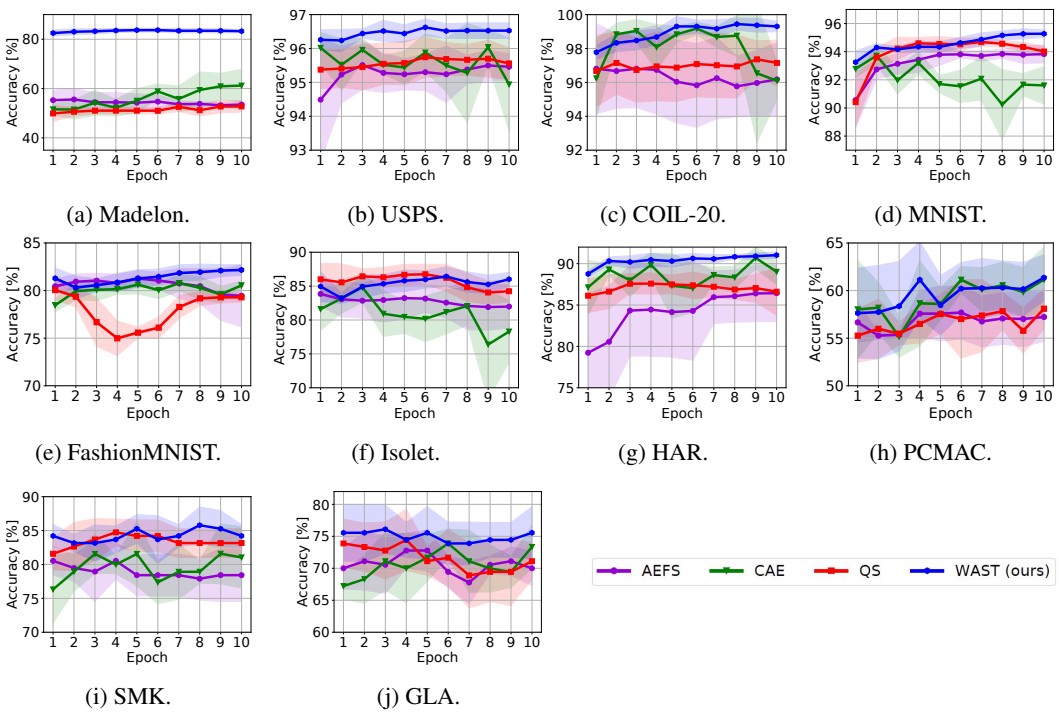

Figure 5: The performance of different methods at the early stages of the training. The test accuracy is reported for at first 10 training epochs using $K$ of 50 for all datasets except Madelon, where $K$ of 20 is used.

## 5 Analysis

### 5.1 Effect of Fast Attention During Training

To analyze the impact of attention to the informative features in WAST during training, we study the learning behavior of all unsupervised NN-based methods on the test data for the first 10 epochs. Since CAE has a parameter that is dependent on the max number of epochs, we performed ten separate runs and varied the max number of epochs each time. Figure 5 shows the accuracy using $K$ of 50 for all datasets except Madelon, where $K$ of 20 is used. The analysis reveals the following findings: **(1) Robustness to noisy environments:** The results on Madelon illustrate the robustness of WAST against the noisy features (480 out of 500). The figure shows the performance gap between WAST and the baseline methods starting from epoch 1. After 10 epochs, WAST outperforms the second-best performer by 22%. More experiments on noisy enviroments can be found in Appendix G. **(2) Performance on datasets with a large number of samples:** WAST outperforms CAE by 3.6% and 1.6% on MNIST and FashionMNIST, respectively, although CAE was the best performer on these datasets after training it for a larger number of epochs (Table 2). **(3) Consistency across different dataset types and characteristics:** We observe that some baselines achieve high performance in some domains while being worse in others. WAST has the highest consistency on different dataset types and characteristics, outperforming all baselines after 10 epochs (Appendix C). **(4) Stability:** WAST is more stable at the early stage of the training. This is represented by the standard deviation across different training runs (shaded region).

### 5.2 Visualization

Figure 6 illustrates how the sparse topology changes during training in WAST and QS. We performed this analysis on MNIST, where handwritten digits are centered in $28 \times 28$ grayscale images (Figure 6a). Initially, the sparse connections are uniformly distributed on the input neurons. With the guided adaptation of a sparse topology via attention, WAST detects the informative features quickly after one

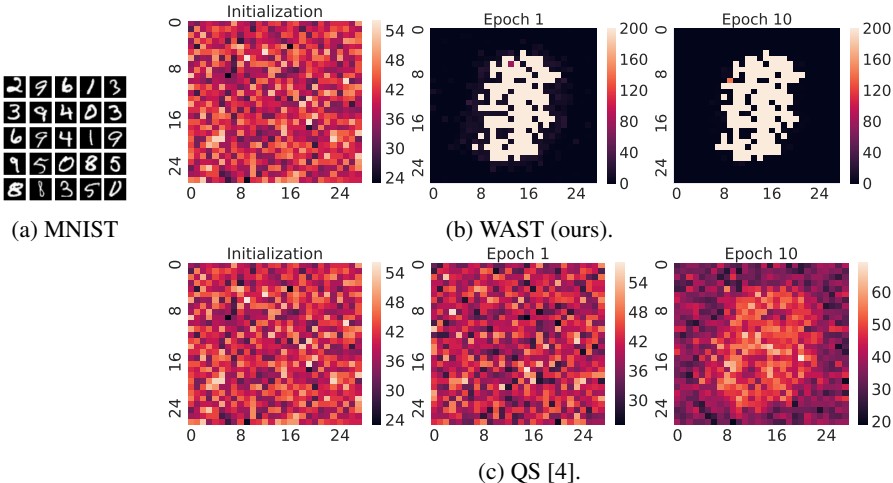

(a) MNIST

(b) WAST (ours).

(c) QS [4].

Figure 6: Visualization. (a) Samples of the MNIST dataset. Each sample is $28 \times 28$, and the informative pixels are centered in the middle. The distribution of the sparse connections during training (i.e., the number of outgoing connections from each input neuron) on MNIST using WAST (b) and the QS method [4] (c). WAST has faster attention to the informative features after one epoch.

Table 4: Ablation study. Effect of each component in the criteria of neuron importance and connection importance. The accuracy (%) is reported using $K$ of 20 and 50 on Madelon and others, respectively.

| Criteria | # | Method | Madelon | USPS | HAR | PCMAC | SMK |
|---|---|---|---|---|---|---|---|
| Neuron | 1 | "w/o $\lvert\frac{\partial L}{\partial \bar{\mathbf{x}}}\rvert$" | 55.50±2.45 | 95.91±0.64 | 87.88±1.79 | 55.58±2.78 | 84.21±3.16 |
| Neuron | 2 | "w/o \|**W**\|" | 82.47±0.75 | 95.56±0.18 | **92.20±0.50** | 58.66±4.18 | 75.26±2.11 |
| Neuron | 3 | "w/o momentum" | 81.60±1.06 | 96.60±0.40 | 90.74±0.94 | 58.30±4.80 | 78.95±3.33 |
| Connection | 4 | "w/o $\mathcal{I}_i$" | 83.27±0.63 | 96.54±0.17 | 90.11±0.52 | 55.53±3.07 | 84.74±1.05 |
| | | WAST (ours) | **83.27±0.63** | **96.69±0.27** | 91.20±0.20 | **60.51±2.53** | **84.74±1.05** |

epoch. On the other hand, the random growth of the connections in QS requires more training epochs. The effect of increasing the frequency of topological exploration in QS is studied in Appendix F.

## 5.3 Ablation Study

We performed an ablation study to assess the contribution of each component in the proposed criteria of neuron and connection importance in the performance of WAST.

**Neuron Importance.** (#1) "w/o $\lvert\frac{\partial L}{\partial \bar{\mathbf{x}}}\rvert$": Using only the magnitude of connected weights to determine the importance of a neuron (i.e., $\lambda = 0$), (#2) "w/o |**W**|": Using only the sensitivity of the neuron to the change in the loss (i.e., $\lambda = 1$), and (#3) "w/o momentum": The importance of a neuron is based only on the *current* estimate of its sensitivity to the loss and currently connected weights.

**Connection Importance.** (#4) "w/o $\mathcal{I}_i$": The importance of a connection is estimated by its magnitude without considering the importance of its connected neuron $\mathcal{I}_i$.

We performed this analysis on datasets of different types. The results are summarized in Table 4. We observe that removing the sensitivity of a neuron to the loss (#1) has the biggest effect on performance. It has the highest impact on noisy environments, where it reduces the performance by 27.77% on Madelon. On the other datasets, it results in a $0.53 - 4.93\%$ reduction in the performance. Excluding the magnitude of the weights (#2) affects mainly the datasets with few training samples (e.g., SMK). Neglecting the neuron importance in the previous training iterations (#3) reduces the performance by $0.09 - 5.79\%$. Aligned with many prior works [20; 54; 27], we find that the weight magnitude is an effective metric for estimating the connection importance (#4). Moreover, our results show that enhancing it with the importance of its connected neuron improves the performance even further on some datasets without diminishing it on the others.

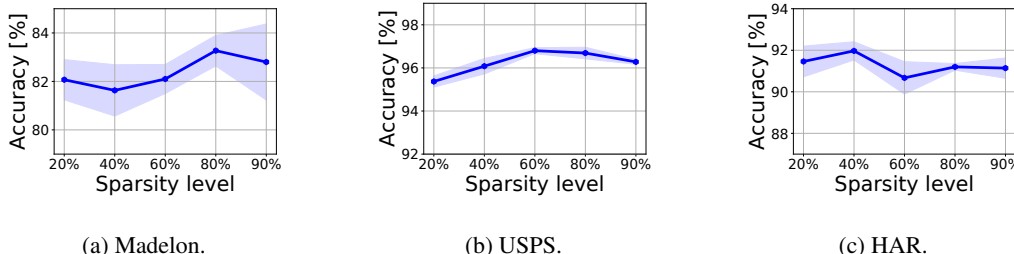

(a) Madelon.                    (b) USPS.                    (c) HAR.

Figure 7: The performance of WAST using different sparsity levels for the autoencoder model. The test accuracy is reported using $K$ of 50 except on Madelon, where $K = 20$.

## 5.4 Effect of the Sparsity Level

We further study the effect of the sparsity level of the auto-encoder model on the performance of WAST. We evaluated 5 different sparsity levels $\{20\%, 40\%, 60\%, 80\%, 90\%\}$. All other settings are the same as the ones stated in Appendix A.

Figure 7 illustrates the classification accuracy using $K = 20$ for Madelon and K=50 for other datasets. We observe the performance of WAST is robust to the sparsity level. Yet, we mostly care about the performance at high sparsity levels, as the goal is to achieve high performance at a low computational cost.

## 6 Discussion

**Conclusion.** In this paper, we propose WAST, a new efficient neural network-based method for unsupervised feature selection. We train a sparse autoencoder from scratch and optimize the sparse topology during training to detect the informative features quickly. We performed extensive experiments in which we evaluated 55 cases on various datasets and different values of the number of selected features. WAST achieves the best performance on 19 cases, while the second-best unsupervised performer has a score of 11 cases. More interestingly, the superior performance is achieved with a few training iterations. WAST reduces the memory and computational costs by 80% and 98%, respectively. Moreover, we show the robustness of WAST towards very noisy environments, outperforming the state-of-the-art methods by 22% with limited training iterations. Finally, we show that WAST performs competitively with supervised-based methods, outperforming them on image datasets. This demonstrates the promise of adapting WAST in the supervised setting and motivates new sparse training algorithms for supervised and unsupervised feature selection.

**Limitations.** This work is a step toward exploiting the power of neural networks for feature selection in a computationally efficient manner. Besides the improvement in the accuracy of selecting the informative feature, fast attention to the important features during training reduces the number of training iterations substantially. Yet, like most sparse training methods in the literature, this proof-of-concept does not fully utilize the memory and computational advantages of sparse neural networks. This is due to the lack of hardware support for sparsity and the higher focus of the community on the algorithmic side [27]. Nevertheless, there is recent growing attention to hardware and software support for sparsity [28; 76; 66; 48; 15; 3; 13] (See Appendix H for discussion). This enables the pure sparse implementation of the method in the future.

**Societal Impacts.** With the emergence of big data, systems that can quickly select informative features and remove redundant ones become crucial. Besides the performance gain that could be achieved in the downstream tasks by removing irrelevant features, reducing the number of features improves memory and computational efficiency significantly. This enables providing energy-efficient systems. Moreover, it is useful for improving the interpretability of model-driven decisions. The *unsupervised* selection of the informative features is effective in cases where labeled data is limited or very expensive to collect. We do not expect that there is a negative societal impact.

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
