

(a) COIL-20.

(b) PCMAC.

(c) SMK.

(d) GLA.

Figure 8: The average values during training of the two components used in the criteria for neuron importance in the input layer: the absolute gradient of the loss with respect to the reconstructed samples and the sum of the absolute weights connected to a neuron.

## A    Experimental Settings

### A.1    Implementation Details

For all datasets, we used standard normalization that scales the features to have zero mean and standard deviation of one. The architecture of the autoencoder consists of one hidden layer with sigmoid activation. A linear activation is used for the output layer. We use a hidden layer of 200 neurons for all datasets. We trained each dataset for 10 epochs using stochastic gradient descent with a momentum of 0.9 and a batch size of 128. We used a learning rate of 0.1 for all datasets except for the datasets with very few samples. We used a learning rate of 0.01 and 0.001 for SMK and GLA, respectively. For sparse-based methods, WAST and QS, we used a sparsity level $s$ of 0.8. The factor of dropped and regrown connections, $\alpha$, in the two methods is 0.3. We used $\lambda$ of 0.4 for all datasets of type image, and $\lambda$ of 0.9 for all other types except the biological data. For SMK and GLA, we used $\lambda$ of 0.01 and 0.001, respectively. The selection of the value of $\lambda$ is guided by the magnitude of the gradient of the loss with respect to the reconstructed sample compared to the magnitude of the weights. This is illustrated in Figure 8. We use $\lambda$ to control the balance between the two components of the neuron importance in Equation 3. For instance, for biological data (SMK and GLA), where the absolute gradient is higher than the average sum of absolute weights connected to a neuron, we use a small value for $\lambda$. The other hyperparameters are selected using a random search. For Madelon, we evaluate the performance of $K = 20$. For all other datasets, we test 6 different values $K \in \{25, 50, 75, 100, 150, 200\}$.

### A.2    Evaluation Metrics

**Accuracy.** To assess how informative are the selected features by each method, we train a support vector machine classifier on the selected features and report the classification accuracy.

**Score.** To summarize the results for all datasets and various values of the selected features $K$, we calculate a score for each studied method. The score of a method increases by one when it is the best performer in terms of classification accuracy for a certain dataset and a certain value of $K$.

**Network size (#params).** To estimate the memory cost of a network, the network size is calculated by the summation of the number of connections allocated in its layers as follows:

$$\#params = \sum_{l=1}^{L} \left\| W^{(l)} \right\|_0 , \tag{5}$$

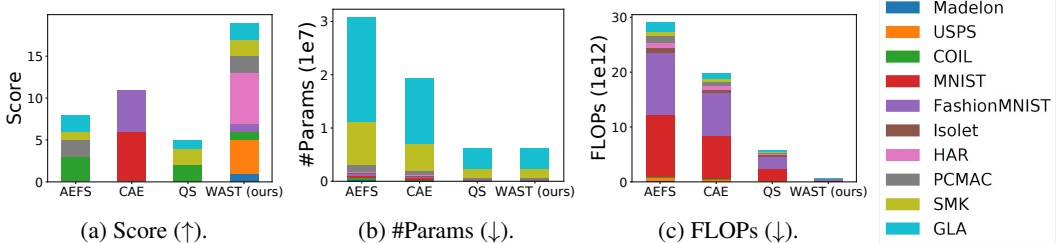

Figure 9: The performance of different methods on the studied datasets across different dimensions: score (a), memory cost (b), and computational cost (c). The score is accumulated for the 6 studied values for $K$.

where $W^{(l)}$ is the weights in layer $l$, $\|.\|_0$ is the standard $L_0$ norm, and $L$ is the number of layers in the model. For sparse neural networks, $\|W^{(l)}\|_0$ is controlled by the defined sparsity level for the model.

**Floating-point operations (FLOPs).** To estimate the computational cost of training a neural network model on a given dataset, we calculate how many FLOPs are performed in the whole course of training. We follow the method described in [20] to calculate the FLOPs. The FLOPs are estimated by the total number of multiplications and additions performed in the forward and backward passes. It is calculated layer by layer in the model and are dependent on the sparsity level of the network.

FLOPs is the typical used metric in the literature to compare the computational cost of a sparse model against its dense counterpart [27]. The main motivation is that current sparse training methods are prototyped using masks over dense weights to simulate sparsity [27]. This is because most deep learning specialized hardware is optimized for dense matrix operations. Therefore, the running time using these prototypes would not reflect the actual gain in memory and speed using a truly sparse network. See Appendix H for further discussion.

**Performance.** To assess the efficiency of a method across different dimensions (accuracy, memory cost, and computational cost) on various types of datasets with different characteristics, we include a comparison that gives a holistic view of all datasets and the studied values for $K$. We study 55 cases ((9 datasets $\times$ 6 values for $K$)+ Madelon dataset). We calculate the total score for all cases and memory and computational costs. Since the memory and computational costs for CAE [5] are dependent on the value of $K$, we report the average costs across the 6 studied values. The memory and computational efficiency are equivalent to $(1-$ cost$)$. Normalized values, using min-max scaling, are illustrated in Figure 3.

## B Additional Experiments

In this appendix, we evaluate the accuracy of the studied unsupervised and supervised methods for various values of the selected features $K \in \{25, 50, 75, 100, 150, 200\}$.

Tables 5 and 6 show the accuracy in each case. Consistent with the observations for $K$ of 50 in Section 4.4, we find that unsupervised neural network-based methods outperform the classical ones in most cases. CAE is the best performer for image datasets with a large number of samples (i.e., MNIST and Fashion MNIST) for all values of $K$ except one. WAST is the best performer in 19 cases, while the second-best performer is the best in 11 cases.

Similarly, the supervised neural network-based methods outperform the classical supervised methods in most cases. Yet, for text and high dimensional datasets, classical methods are competitive with the NN-based ones, outperforming them in 5 values for $K$ on SMK and PCMAC.

Interestingly, WAST outperforms the best supervised performer on the image datasets in 14 cases of 24. On the other types of datasets, supervised methods achieve higher performance in most cases.

Figure 9 summarizes the performance of unsupervised methods across various dimensions. We report the score (i.e., how many times the method is the best performer) on different values of $K$ (Figure 9a), the memory cost estimated by the number of required network parameters (Figure 9b), and the computational cost (Figure 9c). Since the network architecture is dependent on $K$ in CAE [5], we

Table 5: Classification accuracy (%) using unsupervised and supervised feature selection methods for different $K$ selected features. The best performer from the unsupervised methods is in bold font, while the best performer from the supervised methods is in blue.

| Dataset | | | Method/$K$ | 25 | 50 | 75 | 100 | 150 | 200 |
|---|---|---|---|---|---|---|---|---|---|
| USPS | Unsupervised | Classical | lap_score [26] | 63.01±0.00 | 70.54±0.00 | 86.02±0.00 | 90.75±0.00 | 93.87±0.00 | 95.97±0.00 |
| | | | MCFS [9] | 93.49±0.00 | 93.33±0.00 | **97.15±0.00** | **97.53±0.00** | 97.31±0.00 | 97.26±0.00 |
| | | | DUFS [42] | 90.53±1.97 | 95.62±0.54 | 96.37±0.42 | 96.85±0.39 | 97.16±0.23 | 97.49±0.07 |
| | | NN-based | AEFS [25] | 94.04±0.41 | 95.86±0.48 | 96.70±0.17 | 96.86±0.05 | 97.00±0.24 | 96.96±0.22 |
| | | | CAE [5] | 92.50±1.45 | 95.04±0.59 | 95.56±0.26 | 96.10±0.34 | 95.88±0.23 | 96.16±0.05 |
| | | | QS [4] | 93.18±0.72 | 95.88±0.31 | 96.72±0.32 | 97.01±0.13 | 97.41±0.14 | 97.48±0.05 |
| | | | WAST (ours) | **94.59±0.74** | **96.69±0.27** | 97.02±0.10 | 97.11±0.11 | **97.41±0.13** | **97.65±0.09** |
| | Supervised | Classical | Fisher_score [22] | 81.99±0.00 | 91.02±0.00 | 94.35±0.00 | 96.51±0.00 | 97.26±0.00 | 97.53±0.00 |
| | | | L1_L21 [45] | 77.53±0.00 | 86.99±0.00 | 91.51±0.00 | 91.99±0.00 | 95.54±0.00 | 96.83±0.00 |
| | | | CIFE [41] | 50.16±0.00 | 61.29±0.00 | 67.96±0.00 | 78.01±0.00 | 89.57±0.00 | 96.34±0.00 |
| | | | ICAP [33] | 89.95±0.00 | 95.22±0.00 | 95.27±0.00 | 95.38±0.00 | 95.75±0.00 | 96.94±0.00 |
| | | | RFS [56] | 87.37±0.00 | 95.32±0.00 | 96.45±0.00 | 96.72±0.00 | 97.20±0.00 | 97.26±0.00 |
| | | NN-based | LassoNet [37] | 93.56±0.43 | 95.80±0.12 | 96.56±0.09 | 96.98±0.18 | 97.55±0.05 | 97.64±0.04 |
| | | | STG [71] | 93.57±0.32 | 95.78±0.60 | 96.58±0.27 | 97.04±0.13 | 97.37±0.11 | 97.37±0.05 |
| COIL-20 | Unsupervised | Classical | lap_score [26] | 60.42±0.00 | 78.12±0.00 | 80.90±0.00 | 82.29±0.00 | 87.85±0.00 | 88.19±0.00 |
| | | | MCFS [9] | 91.32±0.00 | 97.22±0.00 | 96.53±0.00 | 98.26±0.00 | 98.61±0.00 | 99.65±0.00 |
| | | | DUFS [42] | 92.01±2.91 | 97.43±1.22 | 98.06±1.58 | 98.82±1.53 | 99.79±0.28 | 99.65±0.38 |
| | | NN-based | AEFS [25] | 96.66±1.36 | 99.48±0.41 | 99.02±0.56 | **99.94±0.12** | **100.00±0.0** | **99.94±0.12** |
| | | | CAE [5] | 84.08±2.12 | 94.54±2.92 | 96.74±1.48 | 98.34±1.26 | 97.32±1.05 | 99.88±0.15 |
| | | | QS [4] | **97.29±0.71** | 99.17±0.42 | **99.17±0.47** | 98.89±0.60 | 99.24±0.14 | 99.10±0.28 |
| | | | WAST (ours) | 94.86±1.39 | **99.58±0.14** | 99.03±0.67 | 98.89±0.92 | 99.38±0.40 | 99.51±0.35 |
| | Supervised | Classical | Fisher_score [22] | 50.35±0.00 | 88.89±0.00 | 93.40±0.00 | 95.14±0.00 | 96.18±0.00 | 97.92±0.00 |
| | | | L1_L21 [45] | 90.97±0.00 | 92.01±0.00 | 92.71±0.00 | 92.71±0.00 | 98.26±0.00 | 98.96±0.00 |
| | | | CIFE [41] | 50.69±0.00 | 59.38±0.00 | 63.19±0.00 | 67.71±0.00 | 71.88±0.00 | 72.22±0.00 |
| | | | ICAP [33] | 94.44±0.00 | 99.31±0.00 | 98.96±0.00 | 100.0±0.00 | 100.0±0.00 | 99.31±0.00 |
| | | | RFS [56] | 34.72±0.00 | 66.32±0.00 | 72.22±0.00 | 78.47±0.00 | 86.11±0.00 | 90.28±0.00 |
| | | NN-based | LassoNet [37] | 91.74±0.94 | 95.83±1.18 | 98.89±0.34 | 99.31±0.00 | 99.45±0.28 | 100.0±0.00 |
| | | | STG [71] | 93.40±1.72 | 97.57±1.7 | 99.17±0.87 | 98.89±1.27 | 98.96±1.39 | 99.44±0.72 |
| MNIST | Unsupervised | Classical | lap_score [26] | 14.14±0.00 | 23.94±0.00 | 30.49±0.00 | 40.15±0.00 | 50.99±0.00 | 59.92±0.00 |
| | | | MCFS [9] | - | - | | | | |
| | | | DUFS [42] | 47.23±0.00 | 62.09±0.00 | 69.48±0.00 | 71.27±0.00 | 87.60±0.00 | 88.67±0.00 |
| | | NN-based | AEFS [25] | 87.52±1.47 | 93.22±1.38 | 95.78±0.19 | 96.16±0.50 | 96.72±0.17 | 97.14±0.08 |
| | | | CAE [5] | **91.18±0.68** | **96.20±0.14** | **97.36±0.23** | **97.66±0.08** | **97.90±0.06** | **98.08±0.07** |
| | | | QS [4] | 87.52±0.74 | 94.07±0.40 | 96.00±0.19 | 96.85±0.20 | 97.49±0.07 | 97.88±0.06 |
| | | | WAST (ours) | 88.89±0.88 | 95.27±0.26 | 96.76±0.19 | 97.27±0.04 | 97.69±0.09 | 98.06±0.09 |
| | Supervised | Classical | Fisher_score [22] | 78.50±0.00 | 86.11±0.00 | 91.14±0.00 | 94.42±0.00 | 96.39±0.00 | 97.51±0.00 |
| | | | L1_L21 [45] | 44.64±0.00 | 62.26±0.00 | 75.93±0.00 | 78.27±0.00 | 90.56±0.00 | 90.71±0.00 |
| | | | CIFE [41] | 80.92±0.00 | 89.30±0.00 | 92.75±0.00 | 95.09±0.00 | 96.76±0.00 | 97.62±0.00 |
| | | | ICAP [33] | 81.61±0.00 | 89.03±0.00 | 92.43±0.00 | 94.99±0.00 | 96.44±0.00 | 97.59±0.00 |
| | | | RFS [56] | - | - | - | - | - | - |
| | | NN-based | LassoNet [37] | 86.30±1.07 | 94.38±0.12 | 95.83±0.13 | 96.57±0.07 | 97.43±0.09 | 97.92±0.04 |
| | | | STG [71] | 84.81±0.93 | 92.53±0.86 | 95.56±0.33 | 96.56±0.13 | 97.53±0.08 | 97.99±0.06 |
| Fashion MNIST | Unsupervised | Classical | lap_score [26] | 18.47±0.00 | 27.07±0.00 | 44.16±0.00 | 53.45±0.00 | 69.56±0.00 | 77.32±0.00 |
| | | | MCFS [9] | - | - | | | | |
| | | | DUFS [42] | 45.55±13.86 | 74.69±1.86 | 80.35±1.44 | 83.25±0.67 | 85.46±0.26 | 86.40±0.36 |
| | | NN-based | AEFS [25] | 75.96±1.44 | 80.88±0.71 | 82.66±0.90 | 83.38±0.71 | 84.30±0.67 | 85.42±0.49 |
| | | | CAE [5] | **81.24±0.57** | **84.66±0.16** | **85.74±0.34** | **86.60±0.21** | **86.96±0.22** | 87.46±0.17 |
| | | | QS [4] | 77.96±1.11 | 82.65±0.38 | 84.51±0.47 | 85.45±0.46 | 86.64±0.12 | 87.04±0.09 |
| | | | WAST (ours) | 73.67±1.59 | 82.16±0.57 | 84.42±0.33 | 85.28±0.07 | 86.37±0.25 | **87.58±0.12** |
| | Supervised | Classical | Fisher_score [22] | 53.10±0.00 | 67.85±0.00 | 74.31±0.00 | 79.59±0.00 | 83.55±0.00 | 84.67±0.00 |
| | | | L1_L21 [45] | 63.98±0.00 | 69.57±0.00 | 72.1±0.00 | 72.79±0.00 | 79.08±0.00 | 80.53±0.00 |
| | | | CIFE [41] | 63.36±0.00 | 66.86±0.00 | 67.66±0.00 | 69.18±0.00 | 75.65±0.00 | 78.78±0.00 |
| | | | ICAP [33] | 50.07±0.00 | 59.52±0.00 | 67.22±0.00 | 77.75±0.00 | 81.69±0.00 | 84.53±0.00 |
| | | | RFS [56] | - | - | - | - | - | - |
| | | NN-based | LassoNet [37] | 78.85±0.23 | 82.63±0.23 | 84.03±0.09 | 85.10±0.21 | 86.02±0.13 | 86.51±0.11 |
| | | | STG [71] | 76.41±1.59 | 83.32±0.45 | 85.04±0.29 | 86.05±0.28 | 86.93±0.09 | 87.35±0.11 |

report the average memory and computational costs on different $K$. Datasets with a large number of samples, MNIST and FashionMNIST, cause the highest costs. AEFS requires the highest memory and computational costs. Yet, AEFS performs better than the other dense baseline CAE on various dataset types. QS reduces memory and computational costs substantially. However, it has a lower score than AEFS. WAST achieves the best trade-off across different dimensions. It has the highest score and lowest computational cost.

# C   Effect of Fast Attention During Training

We studied the performance of unsupervised NN-based methods during the first 10 epochs in Section 5.1. In this appendix, we report the accuracy after performing the 10 epochs.

As illustrated in Table 7, on different dataset types (artificial, image, speech, time series, text, and biological) and different characteristics (high/low dimensional features and few-shot/large data), WAST consistently outperforms other NN-based unsupervised methods.

Table 6: Classification accuracy (%) using unsupervised and supervised feature selection methods for different $K$ selected features. The best performer from the unsupervised methods is indicated in bold font, while the best performer from the supervised methods is indicated in blue color.

| Dataset | | | Method/$K$ | 25 | 50 | 75 | 100 | 150 | 200 |
|---|---|---|---|---|---|---|---|---|---|
| Isolet | Unsupervised | Classical | lap_score [26] | 64.81±0.00 | 75.71±0.00 | 79.42±0.00 | 83.85±0.00 | 86.47±0.00 | 90.38±0.00 |
| | | | MCFS [9] | 72.37±0.00 | 81.41±0.00 | 85.83±0.00 | 88.46±0.00 | 91.35±0.00 | 91.79±0.00 |
| | | | DUFS [42] | **73.92±2.57** | **85.62±2.53** | **89.72±2.31** | **92.04±1.50** | **93.60±1.12** | **94.37±0.43** |
| | | NN-based | AEFS [25] | 73.76±1.13 | 80.94±2.02 | 88.04±1.59 | 89.80±0.72 | 91.66±0.73 | 90.76±1.66 |
| | | | CAE [5] | 60.32±5.55 | 78.90±1.24 | 82.00±2.17 | 84.52±1.58 | 86.66±1.24 | 87.24±1.11 |
| | | | QS [4] | 62.77±0.56 | 74.62±2.12 | 82.17±1.40 | 87.31±1.28 | 92.33±0.23 | 93.46±0.34 |
| | | | WAST (ours) | 70.90±1.99 | 85.33±1.39 | 87.60±1.03 | 88.44±0.76 | 90.23±0.19 | 91.46±0.37 |
| | Supervised | Classical | Fisher_score [22] | 68.72±0.00 | 75.64±0.00 | 83.53±0.00 | 86.60±0.00 | 89.42±0.00 | 93.78±0.00 |
| | | | L1_L21 [45] | 48.72±0.00 | 55.90±0.00 | 56.67±0.00 | 66.92±0.00 | 72.31±0.00 | 73.33±0.00 |
| | | | CIFE [41] | 56.03±0.00 | 59.81±0.00 | 74.29±0.00 | 81.22±0.00 | 85.71±0.00 | 87.95±0.00 |
| | | | ICAP [33] | 67.05±0.00 | 75.06±0.00 | 79.68±0.00 | 82.82±0.00 | 89.29±0.00 | 90.26±0.00 |
| | | | RFS [56] | 66.54±0.00 | 77.31±0.00 | 85.06±0.00 | 87.76±0.00 | 92.50±0.00 | 94.87±0.00 |
| | | NN-based | LassoNet [37] | 76.78±0.32 | 85.70±0.38 | 90.49±0.08 | 93.23±0.55 | 95.15±0.06 | 95.67±0.06 |
| | | | STG [71] | 77.37±4.54 | 89.38± 1.19 | 92.43±0.57 | 93.92±0.16 | 94.85±0.09 | 95.74±0.17 |
| HAR | Unsupervised | Classical | lap_score [26] | 80.83±0.00 | 82.80±0.00 | 83.78±0.00 | 84.66±0.00 | 89.48±0.00 | 92.77±0.00 |
| | | | MCFS [9] | 60.10±0.00 | 80.29±0.00 | 84.39±0.00 | 83.78±0.00 | 91.08±0.00 | 91.04±0.00 |
| | | | DUFS [42] | 73.96±5.37 | 86.90±1.06 | 87.91±1.90 | 88.81±1.23 | 92.18±0.70 | 93.34±0.96 |
| | | NN-based | AEFS [25] | 82.28±6.01 | 89.54±0.44 | 90.76±0.49 | 91.56±0.39 | 91.52±0.96 | 93.30±1.29 |
| | | | CAE [5] | 85.70±2.97 | 86.26±2.41 | 89.60±0.62 | 90.16±1.19 | 90.34±0.60 | 91.26±0.92 |
| | | | QS [4] | 78.62±1.15 | 89.68±0.38 | 89.73±1.77 | 89.96±0.58 | 91.68±0.76 | 92.90±0.46 |
| | | | WAST (ours) | **86.37±0.51** | **91.20±0.16** | **91.99±0.20** | **93.97±0.18** | **95.15±0.25** | **94.96±0.49** |
| | Supervised | Classical | Fisher_score [22] | 81.27±0.00 | 83.68±0.00 | 88.84±0.00 | 89.89±0.00 | 93.08±0.00 | 92.30±0.00 |
| | | | L1_L21 [45] | 79.4±0.00 | 81.30±0.00 | 83.31±0.00 | 84.90±0.00 | 91.92±0.00 | 93.76±0.00 |
| | | | CIFE [41] | 80.22±0.00 | 84.15±0.00 | 84.83±0.00 | 85.34±0.00 | 85.92±0.00 | 85.92±0.00 |
| | | | ICAP [33] | 84.46±0.00 | 88.70±0.00 | 89.24±0.00 | 92.06±0.00 | 93.38±0.00 | 93.32±0.00 |
| | | | RFS [56] | 84.22±0.00 | 88.23±0.00 | 88.53±0.00 | 89.92±0.00 | 91.31±0.00 | 92.67±0.00 |
| | | NN-based | LassoNet [37] | 92.69±0.26 | 93.93±0.15 | 94.62±0.15 | 95.04±0.24 | 95.49±0.18 | 95.45±0.10 |
| | | | STG [71] | 86.44±1.45 | 91.75±0.59 | 93.00±0.14 | 93.75±0.37 | 94.12±0.37 | 94.24±0.09 |
| PCMAC | Unsupervised | Classical | lap_score [26] | 49.61±0.00 | 49.87±0.00 | 50.64±0.00 | 55.78±0.00 | 55.53±0.00 | 58.10±0.00 |
| | | | MCFS [9] | 51.67±0.00 | 53.47±0.00 | 53.98±0.00 | 62.72±0.00 | **75.06±0.00** | **72.75±0.00** |
| | | | DUFS [42] | 54.96±2.02 | 57.79±3.18 | 59.90±3.28 | 60.52±1.63 | 65.04±2.59 | 67.92±1.28 |
| | | NN-based | AEFS [25] | 54.14±1.92 | 60.40±2.37 | **63.40±6.19** | **65.10±1.97** | 65.24±3.11 | 67.48±2.89 |
| | | | CAE [5] | 54.30±0.00 | 55.08±0.00 | 56.24±1.86 | 57.12±2.47 | 62.28±4.28 | 66.12±3.04 |
| | | | QS [4] | 54.65±3.25 | 55.78±3.25 | 59.79±1.28 | 61.65±2.53 | 63.86±2.67 | 67.61±4.19 |
| | | | WAST (ours) | **58.77±2.48** | **60.51±2.53** | 62.37±1.56 | 61.59±2.71 | 65.09±3.50 | 67.51±2.31 |
| | Supervised | Classical | Fisher_score [22] | 81.75±0.00 | 86.38±0.00 | 85.60±0.00 | 84.58±0.00 | 84.58±0.00 | 84.83±0.00 |
| | | | L1_L21 [45] | 54.50±0.00 | 53.98±0.00 | 55.27±0.00 | 56.81±0.00 | 59.64±0.00 | 60.41±0.00 |
| | | | CIFE [41] | 77.12±0.00 | 75.84±0.00 | 74.81±0.00 | 72.49±0.00 | 72.49±0.00 | 75.58±0.00 |
| | | | ICAP [33] | 82.78±0.00 | 87.66±0.00 | 87.92±0.00 | 87.40±0.00 | 87.92±0.00 | 88.43±0.00 |
| | | | RFS [56] | 73.78±0.00 | 67.61±0.00 | 72.24±0.00 | 70.95±0.00 | 74.29±0.00 | 73.52±0.00 |
| | | NN-based | LassoNet [37] | 86.79±0.31 | 86.53±1.25 | 85.30±1.19 | 85.35±0.86 | 84.63±0.44 | 84.83±0.92 |
| | | | STG [71] | 54.4±2.36 | 56.04±1.90 | 59.07±2.29 | 62.11±3.31 | 65.09±2.88 | 68.84±2.26 |
| SMK | Unsupervised | Classical | lap_score [26] | **84.21±0.00** | 81.58±0.00 | 84.21±0.00 | 81.58±0.00 | 84.21±0.00 | 84.21±0.00 |
| | | | MCFS [9] | 65.79±0.00 | 78.95±0.00 | 78.95±0.00 | 81.58±0.00 | 78.95±0.00 | 78.95±0.00 |
| | | | DUFS [42] | 79.47±6.32 | 81.05±3.07 | 81.05±3.87 | 83.16±3.57 | 83.16±3.16 | 82.63±2.68 |
| | | NN-based | AEFS [25] | 72.64±3.57 | 79.48±3.07 | 85.80±3.57 | **85.76±3.59** | 81.58±1.68 | 82.62±2.12 |
| | | | CAE [5] | 77.90±3.17 | 78.94±2.37 | 79.48±2.60 | 83.12±3.57 | 84.20±1.64 | 85.78±2.12 |
| | | | QS [4] | 81.05±5.37 | 81.58±3.72 | 81.58±2.88 | 81.05±4.21 | **85.26±2.11** | **86.32±1.97** |
| | | | WAST (ours) | 76.32±4.99 | **84.74±1.05** | **86.84±2.35** | 85.26±2.68 | 84.74±1.05 | 85.79±1.29 |
| | Supervised | Classical | Fisher_score [22] | 68.42±0.00 | 73.68±0.00 | 76.32±0.00 | 78.95±0.00 | 78.95±0.00 | 78.95±0.00 |
| | | | L1_L21 [45] | 78.95±0.00 | 84.21±0.00 | 89.47±0.00 | 84.21±0.00 | 81.58±0.00 | 81.58±0.00 |
| | | | CIFE [41] | 81.58±0.00 | 81.58±0.00 | 76.32±0.00 | 81.58±0.00 | 81.58±0.00 | 78.95±0.00 |
| | | | ICAP [33] | 78.95±0.00 | 73.68±0.00 | 71.05±0.00 | 76.32±0.00 | 71.05±0.00 | 76.32±0.00 |
| | | | RFS [56] | 78.95±0.00 | 76.32±0.00 | 76.32±0.00 | 71.05±0.00 | 71.05±0.00 | 71.05±0.00 |
| | | NN-based | LassoNet [37] | 75.79±6.74 | 77.37±3.57 | 80.00±1.29 | 81.58±2.88 | 80.00±4.88 | 81.05±1.05 |
| | | | STG [71] | 76.32±2.88 | 81.05±5.37 | 80.53±1.29 | 81.58±4.07 | 81.05±3.07 | 84.21±0.00 |
| GLA | Unsupervised | Classical | lap_score [26] | **72.22±0.00** | 66.67±0.00 | 66.67±0.00 | 69.44±0.00 | 69.44±0.00 | 72.22±0.00 |
| | | | MCFS [9] | 69.44±0.00 | 75.00±0.00 | 69.44±0.00 | 72.22±0.00 | 66.67±0.00 | 77.78±0.00 |
| | | | DUFS [42] | 66.66±5.56 | 70.83±1.39 | 73.61±1.39 | 72.22±0.00 | 69.44±2.77 | 70.83±1.39 |
| | | NN-based | AEFS [25] | 66.68±3.51 | 67.76±6.21 | 71.10±3.76 | 71.12±4.51 | **74.44±2.10** | **73.32±2.86** |
| | | | CAE [5] | 65.00±6.25 | 70.56±4.50 | 71.66±3.23 | 72.20±1.77 | 73.88±4.19 | 72.22±3.51 |
| | | | QS [4] | 67.78±5.72 | 68.89±4.78 | 72.22±3.04 | **73.33±3.83** | 72.22±1.76 | 71.67±2.08 |
| | | | WAST (ours) | 66.67±4.65 | **75.56±4.08** | **74.44±3.24** | 71.11±3.77 | 71.11±1.36 | 69.44±3.04 |
| | Supervised | Classical | Fisher_score [22] | 58.33±0.00 | 63.89±0.00 | 66.67±0.00 | 66.67±0.00 | 63.89±0.00 | 61.11±0.00 |
| | | | L1_L21 [45] | 69.44±0.00 | 69.44±0.00 | 72.22±0.00 | 75.00±0.00 | 72.22±0.00 | 72.22±0.00 |
| | | | CIFE [41] | 61.11±0.00 | 58.33±0.00 | 58.33±0.00 | 58.33±0.00 | 72.22±0.00 | 72.22±0.00 |
| | | | ICAP [33] | 69.44±0.00 | 72.22±0.00 | 72.22±0.00 | 69.44±0.00 | 69.44±0.00 | 72.22±0.00 |
| | | | RFS [56] | - | - | - | - | - | - |
| | | NN-based | LassoNet [37] | 75.56±3.24 | 76.67±2.22 | 75.56±1.11 | 75.00±0.00 | 75.56±1.11 | 75.00±0.00 |
| | | | STG [71] | 68.33±8.71 | 71.11±2.83 | 73.33±2.84 | 73.33±2.84 | 73.33±3.77 | 72.22±2.49 |

Table 7: Classification accuracy (%) (↑) using unsupervised NN-based feature selection methods. 50 features are selected for all datasets except Madelon, where 20 features are used. All methods are trained for 10 epochs.

| Method | Madelon | USPS | COIL-20 | MNIST | FashionMNIST |
|---|---|---|---|---|---|
| AEFS [25] | 53.50±1.17 | 95.46±0.39 | 96.18±2.15 | 93.83±0.63 | 79.45±3.26 |
| CAE [5] | 61.20±6.61 | 94.94±1.37 | 96.20±1.10 | 91.60±1.33 | 80.54±0.46 |
| QS [4] | 52.73±2.28 | 95.55±0.46 | 97.15±0.13 | 94.01±0.37 | 79.29±0.40 |
| WAST (ours) | **83.27±0.63** | **96.69±0.27** | **99.58±0.14** | **95.27±0.26** | **82.16±0.57** |

| Method | Isolet | HAR | PCMAC | SMK | GLA |
|---|---|---|---|---|---|
| AEFS [25] | 81.98±3.17 | 86.43±3.40 | 57.23±2.57 | 78.42±3.87 | 70.00±2.72 |
| CAE [5] | 78.30±4.57 | 89.02±1.54 | 61.16±3.45 | 81.04±4.52 | 73.34±3.32 |
| QS [4] | 84.24±1.56 | 86.61±2.83 | 58.09±2.10 | 83.15±2.10 | 71.11±2.83 |
| WAST (ours) | **85.33±1.39** | **91.20±0.16** | **60.51±2.53** | **84.74±1.05** | **75.56±4.08** |

Table 8: Classification accuracy (%) (↑) using unsupervised NN-based feature selection methods on the FMA dataset.

| Method/$K$ | 25 | 50 | 75 |
|---|---|---|---|
| AEFS [25] | 38.50±1.11 | 41.20±1.48 | 42.74±1.32 |
| CAE [5] | 37.06±1.17 | 39.00±0.84 | 39.90±1.32 |
| QS [4] | 41.50±2.29 | 44.65±1.10 | **45.57±0.37** |
| WAST (ours) | **42.13±2.51** | **44.98±1.10** | 45.45±0.48 |

## D   Performance Evaluation on Music Dataset

In this appendix, we perform additional experiments on a recent music dataset, Free Music Archive (FMA) [16; 17]. FMA includes a collection of 106574 audio tracks with artist and album information categorized into 161 genres. We use FMA-small, which consists of 6400 training samples and 800 testing samples with 140 pre-extracted features. We focus on the genre classification task that categorizes the data into eight classes. We use the same setting defined in Appendix A. For WAST, we use $\lambda$ of 0.4. We evaluate the classification accuracy of unsupervised NN-based methods using $K = \{25, 50, 75\}$.

The results are reported in Table 8. Consistent with our results on other benchmarks, WAST outperforms other unsupervised NN-based methods in two cases out of three, with a marginal performance difference in the third case. The performance gain is accompanied by a reduction of the number of epochs by 90%.

## E   Effect of the Frequency of Topology Update

In this section, we analyze the effect of the schedule used to update the sparse topology. In WAST, the topology is updated after each weight update using a batch of the data. We named this schedule "batch update". We compare this schedule with the one used in the QS baseline, in which the sparse topology is updated after each training epoch (i.e., one pass on the whole data). We denote this schedule as "epoch update".

Table 9: Effect of update schedule of the sparse topology in WAST. The accuracy (%) is reported using $K$ of 20 and 50 on Madelon and others, respectively.

| Update schedule | Madelon | USPS | HAR | PCMAC | SMK |
|---|---|---|---|---|---|
| Epoch update | **83.37±0.62** | 96.17±0.30 | 90.55±0.52 | 57.02±3.53 | 84.21±1.29 |
| Batch update | 83.27±0.63 | **96.69±0.27** | **91.20±0.16** | **60.51±2.53** | **84.74±1.05** |

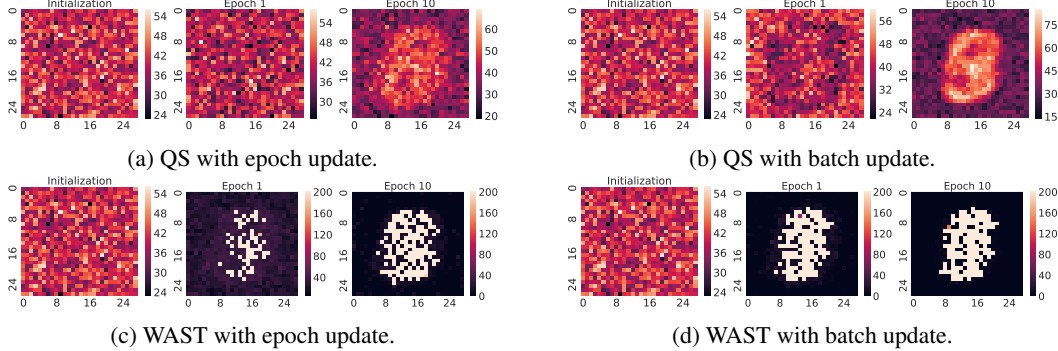

(a) QS with epoch update.

(b) QS with batch update.

(c) WAST with epoch update.

(d) WAST with batch update.

Figure 10: Effect of the update schedule of the sparse topology in the QS baseline and WAST. The sparse topology is updated after each training epoch (left) or after batch update of the weights (right). Updating the topology more frequently increases the speed of detecting the informative features.

Table 9 shows the accuracy on datasets from different types. Updating the topology more frequently by the batch update schedule improves the performance in most of the cases and helps in finding the informative features quicker (Appendix F). The accuracy increases by $0.52 - 3.5\%$ in the studied cases.

## F Visualizing the Effect of the Frequency of Topology Update

In this appendix, we visualize the effect of different update schedules of the sparse topology. We analyzed two schedules: "epoch update" and "batch update" (Appendix E). We performed this analysis on MNIST.

Figure 10 illustrates the effect of different topological update schedules in WAST and QS. The less frequent update of the topology decreases the speed of identifying the informative features. After 1 epoch, the distribution of connections by QS is still almost uniform across different input neurons (Figure 10a). On the other hand, WAST starts to identify the important neurons (Figure 10c). Updating the topology after each batch update increases the speed of identification in the random exploration setting (Figure 10b) and enables very fast detection in WAST after 1 epoch (Figure 10d).

## G Robustness to Noisy Environments

Our experiments on the Madelon dataset illustrate the robustness of WAST to noisy features. In this appendix, we study the robustness of the unsupervised NN-based methods on noisy samples. To this end, we created noisy versions of the datasets by adding Gaussian noise with zero mean to the training data. To assess the robustness of different noise levels, we evaluated 4 different values of standard deviation {0.2, 0.4, 0.6, 0.8}. We performed this analysis on datasets of different types. We use the same experimental setting described in Section 4.3 and Appendix A.

Figure 11 illustrates the classification accuracy using $K = 50$. We observe that across different dataset types, methods that are based on sparse models (i.e., WAST and QS) are more robust to noise. As expected, the accuracy decreases when the level of noise increases in all methods. Yet, sparse-based methods have the least decrease. Moreover, WAST outperforms QS in all cases.

## H Advance in Sparsity

Most of the efforts in previous years are devoted to the algorithmic side of sparse training. The aim is to achieve the same or higher performance than the dense models with highly sparse counterparts, as discussed in Section 2.2.

Existing sparse training methods in the literature simulate sparsity using masks over dense weights since most deep learning specialized hardware is optimized for dense matrix operations. Yet, recently, research on the software and hardware that support sparsity has received growing attention. NVIDIA

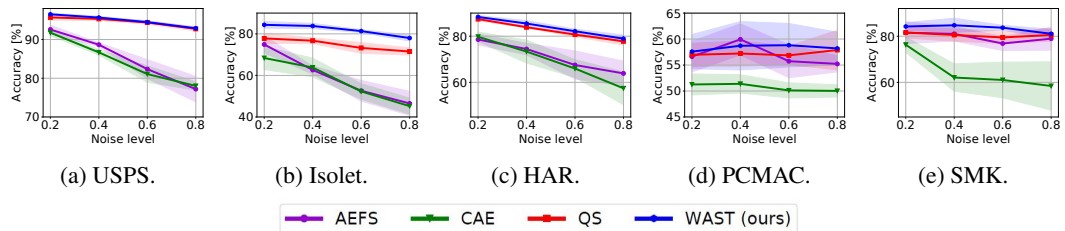

(a) USPS.  (b) Isolet.  (c) HAR.  (d) PCMAC.  (e) SMK.

Figure 11: The performance of different methods on noisy datasets where Gaussian noise is added. We assess the performance on different noise levels by varying the standard deviation of the Gaussian noise. The test accuracy is reported using $K$ of 50.

released NVIDIA A100, which supports a 50% fixed sparsity level [76], and many other efforts on the hardware side are proposed [66; 3; 13; 43]. On the software side, libraries that support truly sparse implementations have been started for supervised learning [48; 15]. With a joint community effort in the algorithmic, software, and hardware directions, we would be able to actually provide faster, memory-efficient, and energy-efficient deep neural networks. Further discussion can be found in [29; 53].