# OpenReview forum: "Where to Pay Attention in Sparse Training for Feature Selection?"
_NeurIPS.cc/2022/Conference — NeurIPS 2022 Accept_

### Official Review · Reviewer_XbZ9 · 2022-07-09

**Rating:** 7
**Confidence:** 4
**Soundness:** 4 excellent
**Presentation:** 3 good
**Contribution:** 3 good

**Summary:**

This paper proposes a novel method for unsupervised feature selection called WAST, which is an adaptation of Quick Selection (Atashgahi et al 2021) to give a more efficient search. The general idea of this and related methods is to pass the features through a simple autoencoder and to keep features which are more important. The main difference between WAST and QS is that, on each epoch, after dropping low-importance weights from the autoencoder, QS regrows random weights, WAST adds weights in a way which takes into account the importance of the weights in the previous epoch.

**Questions:**

QS assigns some importance to the fact that they use denoising autoencoders (because adding noise is supposed to encourage the network to find the most important features), but WAST does not appear to add any noise. Do you have any comment on this? What does this tell us about the approach and why it works?

There is also something quite disturbing about the fact that this method is competitive with supervised methods, given that their objectives are so different. What does this tell us about the approach and why it works?

**Limitations:**

The discussion seems well thought out.

**Strengths And Weaknesses:**

The paper demonstrates good performance (for K=50) on a large number of benchmarks, often beating supervised baselines. More importantly, the paper demonstrates much better performance than *comparable* benchmarks over many values of K, including QS. The paper also shows that WAST is slightly more efficient than QS (which is already much more efficient than some other related methods).

The benchmarks in question are evaluated on whether or not they retain accuracy. However, typical supervised variable selection is evaluated on whether it selects the *right* features, on artificial datasets for which the true features are known. While one could argue that the objectives of unsupervised feature selection are not the same, I think it is quite reasonable, given that the method presented here is competitive with supervised methods, to ask whether it can also go farther than just picking out useful features to picking the correct ones.

The technical advance is somewhat minor, given how close the algorithm is to QS, but on the other hand, the gains are excellent.

---

> ### Author Response · Authors · 2022-08-02
> **Response to Reviewer XbZ9**
>
> We thank the reviewer for the positive and thoughtful reviews and for acknowledging the gains of our method. Please find the answers to the questions below.
>
> **Q1: “QS assigns some importance to the fact that they use denoising autoencoders (because adding noise is supposed to encourage the network to find the most important features), but WAST does not appear to add any noise. Do you have any comment on this? What does this tell us about the approach and why it works?”**
>
> **A1:** We did follow QS in adding noise to the input features. Kindly find this information in Appendix A.1 Lines 530-531.
>
> **Q2: “There is also something quite disturbing about the fact that this method is competitive with supervised methods, given that their objectives are so different. What does this tell us about the approach and why it works?”**
>
> **A2:** This is true that the joint distribution of data and labels can be such that supervised feature selection can lead to much better downstream (classification) results. We can observe this in our experiments too, e.g. the PCMAC dataset. The same holds for other forms of dimensionality reduction / feature extraction and representation learning. However, it is not unreasonable to expect that the compact representation that an autoencoder learns [1] provides us with a good choice of informative features to pick such that they allow the classifier to distinguish different classes of the data. And on some datasets, such unsupervised feature selection may result even in a better subset of features for the downstream classification task. Such behavior was also observed [2] for supervised (e.g., LDA-based) vs. unsupervised (e.g. PCA-based) feature extraction. We hope this explanation clarifies that the results are not disturbing, but are in line with what we expect.
>
> [1] Bengio, Y.: Learning deep architectures for AI. Now Publishers Inc (2009)
>
> [2] Fukunaga, K. Introduction to Statistical Pattern Recognition (1990)
>
> **Other Points:**
>
> **P1: “....to ask whether it can also go farther than just picking out useful features to picking the correct ones”**
>
> **A1:** Thanks for the thoughtful question. We believe that our evaluation on the Madelon dataset could cover this case, where we have only 20 real features and the rest 480 features are purely random. We evaluate the accuracy using only 4% of the features (K=20 features). High accuracy could reflect the feature selection algorithm is able to select the correct ones.

---

> > ### Comment · Reviewer_XbZ9 · 2022-08-09
> > **Response**
> >
> > Thanks for your reply,
> >
> > > We did follow QS in adding noise to the input features. Kindly find this information in Appendix A.1
> >
> > Thanks for the clarification.
> >
> > > on some datasets, such unsupervised feature selection may result even in a better subset of features for the downstream classification task
> >
> > I don't agree that this is the "expected" behaviour, certainly not on underdetermined high-dimensional tasks (e.g. speech tasks, brain-computer interface), but the point is well taken that the fact that unsupervised methods beat supervised ones at picking features for classification is not necessarily aberrant, and not worth mentioning in the paper as a result.
> >
> > > We believe that our evaluation on the Madelon dataset could cover this case, where we have only 20 real features and the rest 480 features are purely random. High accuracy could reflect the feature selection algorithm is able to select the correct ones.
> >
> > It could, and it presumably does, but it is worth checking how well the features actually do overlap with the correct ones.

---

> > > ### Author Response · Authors · 2022-08-09
> > > **Response**
> > >
> > > Thank you. We appreciate the reviewer’s time in checking our response and the constructive suggestions. We will illustrate the overlapping features on Madelon in the final version of the paper.

---

### Official Review · Reviewer_y8xW · 2022-07-09

**Rating:** 8
**Confidence:** 4
**Soundness:** 3 good
**Presentation:** 4 excellent
**Contribution:** 4 excellent

**Summary:**

This paper presents an effective unsupervised method on feature selection via auto-encoders, inspired by the attention mechanism. It acquires the idea of the attention framework and applies it onto the weight-zeroing scheme of the autoencoder recursively, resulting in better training efficiency and promising performance on multiple tasks.

**Questions:**

1. In equation 3), how did you select the \lamda hyper parameter to balance the two components?

2. In the network propagation, is the output processed with a softmax at the end?

3. What do you think on your work's difference with [Dropconnect](https://proceedings.mlr.press/v28/wan13.html)? I know there definitely are some, but as a popular method, you may want to address it.

**Limitations:**

I think the authors have adequately addressed the limitations and potential societal impact of their work.

**Strengths And Weaknesses:**

## Strengths
1. This paper clearly addresses its difference from earlier state-of-the-art level work and identifies its main contributions clearly.
2. The presentation is very clear, with solid statements and references. At least I did not spot any typo or grammatical mistakes there.
3. The novelty is clearly confirmed and proven in the paper, with rigorous mathematical proof.
4. This paper clearly addresses the potential limitation and societal impact.

## Weaknesses
1. The dataset used for benchmarking is a bit out-dated. For example, for spoken digit recognition task, we may want to replace it with a little bit more popular and up-to-date sets such as TIMIT and [yesno](https://www.openslr.org/1/). They are not necessarily harder than the one covered in the paper, but clearly has more work pieces at state-of-the-art level.
2. Also, instead of spoken digit recognition, voice command recognition might be a better choice as tasks, as this paper addresses a lot on real-time application and usage.
3. The work lacks comparison with ad-hoc/hand-crafted features and task-specific features. For example, for spoken digit recognition, one additional experiment with common acoustic features such as [MFCC](https://en.wikipedia.org/wiki/Mel-frequency_cepstrum) may help. But this is trivial.

---

> ### Author Response · Authors · 2022-08-02
> **Response to Reviewer y8xW**
>
> We thank the reviewer for the positive and thoughtful reviews and for acknowledging the novelty of our method. Please find the answers to the questions below.
>
> **Q1:"In equation 3), how did you select the \lamda hyper parameter to balance the two components?"**
>
> **A1:** The search space of the hyperparameter \lamda is guided by analyzing the absolute value of the two components ( the magnitude of the gradient and the sum of absolute weights) for a few iterations. For instance, for SMK, the average value of the magnitude of the gradients is approximately ~0.8, while the average sum of the absolute weight is ~ 0.1. In this case, we choose a small value of  \lamda (i.e., 0.01).
>
> **Q2:” In the network propagation, is the output processed with a softmax at the end?”**
>
> **A2:** We use linear activation for the output layer. Kindly find all architectural details in Appendix A.1.
>
> **Q3: “What do you think on your work's difference with Dropconnect? I know there definitely are some, but as a popular method, you may want to address it.”**
>
> **A3:** One fundamental difference is that DropConnect acts on dense neural networks, while DST methods act on sparse neural networks. DropConnect resembles (broadly speaking) DropOut at the connection levels and aims to act as a regularizer for training a dense neural network. DST-based methods (including our proposed method, WAST) aim to co-optimize during training the values of the weights and the connectivity of sparse neural networks in order to find sparse optimal connectivity capable of good performance. Also, there is a difference at the algorithmic level. While DropConnect inactivates at random a different set of connections at every iteration during training, DST methods remove or add permanently connections during training based on various criteria. Nevertheless, the principle of adding and removing connections (even permanently) also enables some innate regularization capabilities to the DST methods, and this reflects in their competitive performance with dense neural network training.
>
> **Other points:**
>
> **P1: “...for spoken digit recognition task, we may want to replace it with a little bit more popular and up-to-date sets”**
>
> **A1:** Thank you for the suggestion. We performed additional experiments on a recent and popular music dataset Free Music Archive (FMA) [1,2]. FMA includes a collection of 106574 audio tracks with artist and album information categorized in 161 genres. We focus on the genre classification task that categorizes the data into eight classes. The number of input pre-extracted features is 140. We evaluate the accuracy using K={25, 50, 75}. We use the same setting as other benchmarks, where WAST is trained for 10 epochs, while the others are trained for 100 epochs. As shown in the table below, consistent with our results on other datasets, WAST outperforms other unsupervised NN-based methods in most cases.  Please check Appendix D in the revised version of the paper for full details.
>
> |                 | **25**     | **50**      | **75**     |
> |-----------------|------------|-------------|------------|
> | **AEFS**        | 38.50±1.11 | 41.20±1.48  | 42.74±1.32 |
> | **CAE**         | 37.06±1.17 | 39.00±0.84  | 39.90±1.32 |
> | **QS**          | 41.50±2.29 | 44.65±1.10  | 45.57±0.37 |
> | **WAST (ours)** | 42.13±2.51 | 44.98±1.10  | 45.45±0.48 |
>
> [1] Defferrard, M., Benzi, K., Vandergheynst, P., & Bresson, X. (2016). FMA: A dataset for music analysis. arXiv preprint arXiv:1612.01840.
>
> [2] Defferrard, M., Mohanty, S. P., Carroll, S. F., & Salathé, M. (2018, April). Learning to recognize musical genre from audio: Challenge overview. In Companion proceedings of the the web conference 2018 (pp. 1921-1922).

---

### Official Review · Reviewer_cbLE · 2022-07-10

**Rating:** 6
**Confidence:** 4
**Soundness:** 3 good
**Presentation:** 4 excellent
**Contribution:** 2 fair

**Summary:**

The authors propose an extension to a recent auto-encoder-based sparse network approach for feature selection/extraction. Specifically, they extend the previously proposed drop and grow approach for inducing sparsity by leveraging the contribution of an input or output neuron to the reconstructed output. Experiments on 10 different datasets show that the proposed approach provides statistically significant gains in computational cost compared to the previous approach(es).

**Questions:**

1. Beyond the classical classification tasks, how would the proposed approach fare against the state-of-the-art in CV, NLP, and graph ML tasks?
2. Why are only the input and output neurons subject to the drop and grow process but not the hidden neurons?
3. I can understand the improvements in efficiency but why does the classification accuracy improve with the use of WAST? Section 5.1 describes this additionally but doesn't explain the root causes.


**Limitations:**

Yes, the authors have addressed many of the limitations and societal impact. They might additionally wish to consider the questions above.

**Strengths And Weaknesses:**

Strengths
1. Easy to read and follow paper.
2. Lucid experimental details and crisp presentation.
3. Fair coverage of competing classical approaches.
4. Offer to make the code publicly available.

Growth opportunities
1. The work is an incremental improvement over the previous approach. The proposed approach is intuitively motivated with several judgement calls (on leveraging neuronal weights differently) that are not theoretically justified.
2. The field is moving away from explicit feature selection. Hence, the impact of this work might be limited.

---

> ### Author Response · Authors · 2022-08-02
> **Response to Reviewer cbLE**
>
> We thank the reviewer for the thoughtful feedback and for acknowledging the significance of our method. Please find the answers to the questions below.
>
> **Q1: Beyond the classical classification tasks, how would the proposed approach fare against the state-of-the-art in CV, NLP, and graph ML tasks?**
>
> **A1:** Thank you for this interesting question! We would like to clarify that we use classification accuracy as one of the commonly used techniques for the evaluation purpose of how representative the selected subset of features is [1]. Studying the actual utility of feature selection in concrete applications goes beyond the scope of this paper. It can indeed vary depending on the primary goal of feature selection, including e.g., dimensionality reduction, removal of irrelevant and/or redundant features, knowledge discovery, interpretability, supervised (e.g., classification), unsupervised (e.g., clustering) or reinforcement learning, and on the application domain including CV, NLP and graph analytics. Studying the utility of our approach for downstream classification tasks would be indeed the most straightforward, but not the most obvious in the case of CV, NLP, and graph ML - since those tasks would commonly require feature extraction step(s) to make the application of feature selection more meaningful [2]. If we can setup and run enough experiments in the coming period, we will add a statement about the potential of our approach.
>
> [1] Cai, J., Luo, J., Wang, S., Yang, S.: Feature selection in machine learning: A new perspective. 327 Neurocomputing 300, 70–79 (2018)
>
> [2] Bolon-Canedo, V., & Remeseiro, B. (2020). Feature selection in image analysis: a survey. Artificial Intelligence Review, 53(4), 2905-2931.
>
> **Q2: “Why are only the input and output neurons subject to the drop and grow process but not the hidden neurons?”**
>
> **A2:** The drop and grow process is performed on the level of the connections. Thus, the distribution of the connections on the hidden neurons is also changing during training. The growth of the connections on the input/output neurons is based on its contribution to the loss to pay fast attention to informative input features, while the probability of growth in a hidden neuron is uniform since we consider the hidden neuron equally important. We briefly experimented, during the development of the approach, estimating the importance of each neuron based on the gradient of its connections with respect to the loss and observed that the performance is comparable to treating the hidden neuron equally important. Yet, it would be an interesting future work to design new criteria for estimating the importance of hidden neurons to increase the learning speed of the hidden representation and consequently detect the informative features even faster.
>
> **Q3:” I can understand the improvements in efficiency but why does the classification accuracy improve with the use of WAST? Section 5.1 describes this additionally but doesn't explain the root causes.”**
>
> **A3:** The classification accuracy is a factor of the chosen K features. The more informative the selected features are, the higher the classification accuracy is. WAST is able to select more informative features based on the attention mechanism used during training.

---

### Official Review · Reviewer_U347 · 2022-07-10

**Rating:** 6
**Confidence:** 5
**Soundness:** 3 good
**Presentation:** 4 excellent
**Contribution:** 2 fair

**Summary:**

The authors propose a method for unsupervised feature selection which is based on Dynamic Sparse Training applied to an autoencoder with a single hidden dimension and trained with reconstruction loss. The number of selected features is given as an hyperparameter along with sparsity level of the NN. Fixing the number of epochs to be 10 for a proposed method while the other methods are trained with 100 epochs, produces comparable results in terms of accuracy to both supervised and unsupervised methods.

**Questions:**

* Could you please provide a more detailed comparison to the QS method. Although your method outperforms it, the improvement is marginal in 7 out of 10 datasets (in t.o. accuracy)?
* Is it a typo in Table 2 (AEFS has the maximal accuracy on FashinMNIST)?
* Have you re-implemented other methods by yourself or used them as-is? I didn’t find them in your supplementary.
* Line 230 (Consistency and Stability) - I think it is better to provide a numeric comparison to support your claim.
* Why are some results missing from Table 2?
* Once you compare your method to classical unsupervised FS, I think it could be interesting to compare the total training time between non-* NNet and NNet-based methods.
* Have you conducted experiments on sparsity level hyper parameter, how does it affects the performance of your method (accuracy/sparsity tradeoff)?


**Limitations:**

The limitations are mentioned, also the social impact.

**Strengths And Weaknesses:**

Strengths:
+ The method reduces the number of epochs x10 less than other methods
+ The experiments conducted on datasets that come from different domains: image, speech, time series, text, bio
+ Simple method, with clear description, easy to follow
+ Single loss term without additional reguralizations
+ Outperforms almost in all datasets selected for the experiments

Weaknesses:
 - Some related recent works are missed, f.e. DUFS [1] - the method outperforms both CAE and MCFS on real datasets
 - The author has a claim on “robustness of our method in extremely noisy environments and its effectiveness for datasets with very high dimensional feature space and a few training samples”, which is misleading. Noisy environment is defined both in terms of the number of noise features which are unrelated to dataset labels, and also in terms of noise in data samples. This work doesn’t provide any experimental results on datasets with noisy samples.
- The datasets that were chosen to validate the effectiveness of the method are not challenging since the number of samples in the datasets larger or near the same as the number of features.
- The supervised methods chosen for comparison are mostly classical non-NNet methods so they couldn’t serve as an upper bound (in terms of accuracy) for the proposed unsupervised method, while only a single NN-based method was chosen.
- The number of target selected features should be provided as a hyperparameter.
- The experiments run only 5 times for each method and dataset, I think it should be at leaset 10 times.

[1] Lindenbaum, Ofir, et al. "Differentiable unsupervised feature selection based on a gated laplacian." Advances in Neural Information Processing Systems 34 (2021): 1530-1542.

---

> ### Author Response · Authors · 2022-08-02
> **Response to Reviewer U347 (1/4)**
>
> We thank the reviewer for the constructive feedback and positive comments. Below we answer all questions and provide the requested additional experiments and analyses.
>
> **Q1: “Could you please provide a more detailed comparison to the QS method. Although your method outperforms it, the improvement is marginal in 7 out of 10 datasets (in t.o. accuracy)?”**
>
> **A1:** Please find the difference between QS and WAST in various aspects below:
>
> 1. **[Technical aspects]** (a) Unlike QS, which randomly explores different topologies during training, the core idea of WAST is to optimize the sparse topology faster by paying attention to the informative features during training. We provide a visualization example in Figure 6.
>
> &nbsp;&nbsp;&nbsp;&nbsp;&nbsp;&nbsp;(b) The criteria used for the drop and grow phases of the sparse topology adaptation are different in the two algorithms. In summary, QS uses weight magnitude-based drop and random growth, while WAST exploits the neuron importance in the drop and grow mechanisms. We provide this explanation in lines 80-83 in the related work section for QS and Section 3 for WAST.
>
> &nbsp;&nbsp;&nbsp;&nbsp;&nbsp;&nbsp;(c) The update schedule of the sparse topology is different. The effect of the update schedule is analyzed in Appendix E, with visualization in Appendix F.
>
> 2. **[Accuracy]** WAST provides rather a substantial improvement over QS in terms of accuracy.  With 90% less number of training epochs (Table 2),  WAST outperforms QS in 9 benchmarks of out 10 with improvements ranging from 0.4% to 10.71%, where 5 cases have improvements larger than 1.5%.
>
> Using a small number of epochs (i.e., 10) for the two methods (Figure 5), WAST outperforms QS in the 10 benchmarks, with improvements ranging from 1.09% to 30.54%. The improvement per dataset is as follows:
>
> | Artificial | Image |       |       |        | Speech | Time series | Text  | Biological |       |
> |------------|:-----:|:-----:|:-----:|:------:|--------|-------------|-------|------------|-------|
> | Madelon    | USPS  | COIL  | MNIST | FMNIST | Isolet | HAR         | PCMAC | SMK        | GLA   |
> | 30.54%     | 1.14% | 2.43% | 1.26% | 2.87%  | 1.09%  | 4.59%       | 2.42% | 1.59%      | 4.45% |
>
> 3. **[Computational costs]** WAST reduces the number of training epochs by 90%, thanks to the fast attention to informative features during training. Hence, the computational costs are reduced by 90% (Table 3).
>
> **Q2: “Is it a typo in Table 2 (AEFS has the maximal accuracy on FashinMNIST)?”**
>
> **A2:** Thank you for pointing out this. This was a typo indeed. The accuracy of AEFS on FashionMNIST is 80.88±0.71 (as correctly reported in Table 5). Hence, CAE has the best accuracy, as presented in Table 2. We have corrected this in the revised version.
>
> **Q3: “Have you re-implemented other methods by yourself or used them as-is? I didn’t find them in your supplementary.”**
>
> **A3:** We implemented QS. For other methods, we use their official codes and the Scikit-Feature library. We state the details of the implementation used for every method in the main manuscript. We kindly ask the reviewer to check Section 4.3: Experimental Settings “Implementation” Lines 176-182.
>
> **Q4: “Line 230 (Consistency and Stability) - I think it is better to provide a numeric comparison to support your claim.”**
>
> **A4:** Thank you for the suggestion. Please find the numeric comparison below:
>
> **(1) Consistency**
> As shown in the table, on different dataset types (artificial, image, speech, time series, text, biological) and different characteristics (high/low dimensional features, few-shot/large data), WAST consistently outperforms other methods. We included the numeric comparison in Appendix C.
>
> | Method |   Madelon  |    USPS    |    COIL    |    MNIST   |   FMNIST   |
> |:------:|:----------:|:----------:|:----------:|:----------:|:----------:|
> |  AEFS | 53.50±1.17 | 95.46±0.39 | 96.18±2.15 | 93.83±0.63 | 79.45±3.26 |
> |   CAE  | 61.20±6.61 | 94.94±1.37 | 96.20±1.10 | 91.60±1.33 | 80.54±0.46 |
> |   QS   | 52.73±2.28 | 95.55±0.46 | 97.15±0.13 | 94.01±0.37 | 79.29±0.49 |
> |  WAST  | **83.27±0.63** | **96.69±0.27** | **99.58±0.14** | **95.27±0.26** | **82.16±0.57** |
> | **Method** |   **Isolet**   |     **HAR**    |    **PCMAC**   |     **SMK**    |     **GLA**    |
> |  AEFS  | 81.98±3.17 | 86.43±3.40 | 57.23±2.57 | 78.42±3.87 | 70.00±2.72 |
> | CAE    | 78.30±4.57 | 89.02±1.54 | 61.16±3.45 | 81.04±4.52 | 73.34±3.32 |
> | QS     | 84.24±1.56 | 86.61±2.83 | 58.09±2.10 | 83.15±2.10 | 71.11±2.83 |
> | WAST   | **85.33±1.39** | **91.20±0.16** | **60.51±2.53** | **84.74±1.05** | **75.56±4.08** |

---

> > ### Author Response · Authors · 2022-08-02
> > **Response to Reviewer U347 (2/4)**
> >
> > **(2) Stability**
> >
> > For clear readability of the stability during the early stages of training, we believe that it is better to represent it graphically as in Figure 5 instead of reporting 400 numbers (4 methods, 10 epochs, 10 datasets). As an example, we provide the stability, estimated by the standard deviation, at epoch 3 below:
> >
> > | **Method** | **Madelon** | **COIL** | **USPS** | **MNIST** | **FMNIST** | **Method** | **Isolet** | **HAR** | **PCMAC** | **SMK** | **GLA** |
> > |------------|-------------|----------|----------|-----------|------------|------------|------------|---------|-----------|---------|---------|
> > | **AEFS**   | 3.87        | 2.3      | 0.72     | 0.74      | 0.8        | **AEFS**   | 2.58       | 5.49    | 0.38      | 4.42    | 4.51    |
> > | **CAE**    | 5.16        | 0.68     | 0.34     | 0.96      | 1.15       | **CAE**    | 2.18       | 1.95    | 2.14      | 2.88    | 4.52    |
> > | **QS**     | 1.32        | 1.90     | 0.67     | 0.80      | 2.54       | **QS**     | 1.10       | 1.20    | 1.17      | 3.06    | 4.44    |
> > | **WAST**   | 1.06        | 0.56     | 0.13     | 0.19      | 0.60       | **WAST**   | 1.72       | 0.76    | 4.71      | 2.68    | 3.76    |
> >
> > **Q5: “Why are some results missing from Table 2?”**
> >
> > **A5:** In these cases, prohibitive training time is needed either because of the high dimensionality of the features (SMK) or the high number of training samples (MNIST and FashionMNIST). Experiments that exceed 12 hours limit are not considered. We explain this in Lines 180-182 in the manuscript.
> >
> > **Q6:“Once you compare your method to classical unsupervised FS, I think it could be interesting to compare the total training time between non-NNet and NNet-based methods."**
> >
> > **A6:** Motivated by the success of NN-based methods in outperforming non-NN-based ones, our goal is to address their limitation of being computationally extensive. Hence, we focus on evaluating the reduction of the computational costs obtained by our method compared to the other NN-based ones. As the focus of this paper is on the algorithmic side, we leave its truly sparse implementation for the next future work, where the running time that reflects the actual gain of sparsity could be reported. Kindly refer to Section 6 and Appendix I for a detailed discussion.
> >
> > **Q7: “Have you conducted experiments on sparsity level hyper parameter, how does it affects the performance of your method (accuracy/sparsity tradeoff)?”**
> >
> > **A7:** Thank you for the thoughtful question. We analyzed the performance of WAST on 5 different sparsity levels {0.2, 0.4, 0.6, 0.8, 0.9}. Please find the accuracy reported in the table below. Interestingly, the performance of WAST is robust to the sparsity level of the model. Yet, we mostly care about the performance at high sparsity levels, as the goal is to achieve high performance at a low computational cost. Please check Appendix H in the revised version of the paper for full details.
> >
> > | Dataset/Sparsity | 0.2        | 0.4        | 0.6        | 0.8        | 0.9          |
> > |------------------|------------|------------|------------|------------|--------------|
> > | Madelon          | 82.07±0.83 | 81.63±1.06 | 82.10±0.60 | 83.27±0.63 | 82.80 ± 1.57 |
> > | USPS             | 95.37±0.26 | 96.08±0.36 | 96.80±0.14 | 96.69±0.27 | 96.28 ± 0.11 |
> > | HAR              | 91.46±0.74 | 91.97±0.44 | 90.67±0.78 | 91.20±0.16 | 91.14± 0.49  |
> >
> >
> > **Other points:**
> >
> > **P1: “ Some related recent works are missed, f.e. DUFS [1].....”**
> >
> > **A1:** Thanks for your comment. We conducted a comparison with DUFS using the official implementation of the paper. Below is a summary of the accuracy using K=50 on all datasets except Madelon, where K=20. We included the results for other values of K {25, 75, 100, 150, 200} in Appendix B Tables 5 and 6.
> > | Method   | Madelon    | USPS       | COIL       | MNIST      | FMNIST     |
> > |----------|------------|------------|------------|------------|------------|
> > | DUFS [1] | 52.57±5.50 | 95.62±0.54 | 97.43±1.22 | 62.09±0.0  | 74.69±1.86 |
> > | WAST     | 83.27±0.63 | 96.69±0.27 | 99.58±0.14 | 95.27±0.26 | 82.16±0.57 |
> > | **Method**   | **Isolet**     | **HAR**        | **PCMAC**      | **SMK**        | **GLA**        |
> > | DUFS [1] | 85.62±2.53 | 86.90±1.06 | 57.79±3.18 | 81.05±3.07 | 70.83±1.39 |
> > | WAST     | 85.33±1.39 | 91.20±0.16 | 60.51±2.53 | 84.74±1.05 | 75.56±4.08 |
> >
> > [1] Lindenbaum, Ofir, et al. "Differentiable unsupervised feature selection based on a gated laplacian." Advances in Neural Information Processing Systems 34 (2021): 1530-1542.

---

> > > ### Author Response · Authors · 2022-08-02
> > > **Response to Reviewer U347 (3/4)**
> > >
> > > **P2: “....This work doesn’t provide any experimental results on datasets with noisy samples”**
> > >
> > > A2: Thank you for the thoughtful comment. Following your suggestion, we evaluated the performance of unsupervised NN-based methods on noisy samples by adding a gaussian noise with zero mean and different values for the standard deviation {0.2, 0.4, 0.6, 0.8} on the training data. We performed this analysis on datasets of different types. Below, we report the accuracy using K=50. Interestingly, across different domains, sparse-based methods (QS and WAST) have strong robustness to the noise, especially at high noise levels. Moreover, WAST outperforms QS in all cases. Please check Appendix G in the revised version of the paper for full details.
> > >
> > > |          |            | **USPS**   |            |            |          |            | **Isolet** |            |             |
> > > |----------|------------|------------|------------|------------|----------|------------|------------|------------|-------------|
> > > |          | **0.2**    | **0.4**    | **0.6**    | **0.8**    |          | **0.2**    | **0.4**    | **0.6**    | **0.8**     |
> > > | **AEFS** | 92.58±1.12 | 88.66±0.49 | 82.36±2.44 | 77.20±3.29 | **AEFS** | 74.9±4.01  | 62.62±1.98 | 52.64±4.97 | 46.52± 5.76 |
> > > | **CAE**  | 91.76±0.79 | 86.68±0.70 | 81.08±2.11 | 78.02±1.07 | **CAE**  | 68.38±5.54 | 63.82±4.49 | 52.3±3.72  | 45.22±3.84  |
> > > | **QS**   | 95.70±0.71 | 95.32±0.41 | 94.40±0.43 | 92.74±0.53 | **QS**   | 77.81±2.46 | 76.71±1.38 | 73.24±2.59 | 71.51±1.75  |
> > > | **WAST** | 96.51±0.30 | 95.66±0.34 | 94.48±0.30 | 92.90±0.25 | **WAST** | 84.35±1.59 | 83.79±1.28 | 81.33±0.69 | 78.00±1.49  |
> > >
> > > |          |            | **HAR**    |             |            |          |             | **PCMAC**  |            |            |
> > > |----------|------------|------------|-------------|------------|----------|-------------|------------|------------|------------|
> > > |          | **0.2**    | **0.4**    | **0.6**     | **0.8**    |          | **0.2**     | **0.4**    | **0.6**    | **0.8**    |
> > > | **AEFS** | 78.56±2.38 | 74.44±2.49 |  67.52±6.07 | 63.88±5.33 | **AEFS** | 56.62± 2.96 | 59.94±2.98 | 55.74±3.17 | 55.22±1.54 |
> > > | **CAE**  | 79.74±2.05 | 73.34±4.98 | 65.98±3.84  | 57.42±7.11 | **CAE**  | 51.26±2.02  | 51.38±1.76 | 50.12±1.46 | 50.02±1.17 |
> > > | **QS**   | 87.39±0.70 | 83.86±0.29 | 80.68±0.91  | 77.77±1.60 | **QS**   | 56.97±2.21  | 57.22±2.42 | 56.86±1.79 | 57.89±3.80 |
> > > | **WAST** | 88.33±0.60 | 85.47±1.27 | 82.14±0.93  | 78.93±1.54 | **WAST** | 57.58±3.28  | 58.71±4.66 | 58.82±4.18 | 58.20±3.13 |
> > >
> > > **P3:“The datasets that were chosen to validate the effectiveness of the method are not challenging since the number of samples in the datasets larger or near the same as the number of features.”**
> > >
> > > **A3:** Our experiments do cover the cases where the datasets have **few** samples and **very high** dimensional datasets. For instance:
> > > | **Dataset** | **#Sampels** | **# Features** |
> > > |-------------|--------------|----------------|
> > > | **SMK**     | 187          | 19993          |
> > > | **GLA**     | 180          | 49151          |
> > > We kindly ask the reviewer to check Table 1 for the characteristics of the datasets.
> > >
> > > **P4:”...only a single NN-based method was chosen.“**
> > >
> > > **A4:** We followed the reviewer's suggestion and added another recent supervised NN-based method, STG [2]. Please find the results below using K=50, except Madelon, where K=20. WAST outperforms supervised-based methods in 5 cases, while LassoNet and STG are the best performer in 3 and 2 cases, respectively. We included the results for other values of K in Appendix B Tables 5 and 6.
> > >
> > > |                  |                  | **Madelon** | **USPS**   | **COIL**   | **MNIST**  | **FMNIST** |
> > > |------------------|------------------|-------------|------------|------------|------------|------------|
> > > | **Unsupervised** | **WAST**         | 83.27±0.63  | 96.69±0.27 | 99.58±0.14 | 95.27±0.26 | 82.16±0.57 |
> > > | **Supervised**   | **LassoNet [1]** | 79.50±1.22  | 95.80±0.12 | 95.83±1.18 | 94.38±0.12 | 82.63±0.23 |
> > > | **Supervised**   | **STG [2]**      | 59.53±1.90  | 95.78±0.6  | 97.57±1.7  | 92.53±0.86 | 83.32±0.45 |
> > > |                  |                  | **Isolet**  | **HAR**    | **PCMAC**  | **SMK**    | **GLA**    |
> > > | **Unsupervised** | **WAST**         | 85.33±1.39  | 91.20±0.16 | 60.51±2.53 | 84.74±1.05 | 75.56±4.08 |
> > > | **Supervised**   | **LassoNet [1]** | 85.70±0.38  | 93.93±0.15 | 86.53±1.25 | 77.37±3.57 | 76.67±2.22 |
> > > | **Supervised**   | **STG [2]**      | 89.38±1.19  | 91.75±0.59 | 56.04±1.9  | 81.05±5.37 | 71.11±2.83 |
> > >
> > > [1] Lemhadri, Ismael, Feng Ruan, and Rob Tibshirani. "Lassonet: Neural networks with feature sparsity." In International Conference on Artificial Intelligence and Statistics, pp. 10-18. PMLR, 2021.
> > >
> > > [2] Yamada, Yutaro, Ofir Lindenbaum, Sahand Negahban, and Yuval Kluger. "Feature selection using stochastic gates." In International Conference on Machine Learning, pp. 10648-10659. PMLR, 2020.

---

> > > > ### Author Response · Authors · 2022-08-02
> > > > **Response to Reviewer U347 (4/4)**
> > > >
> > > > **P5:The number of target selected features should be provided as a hyperparameter.**
> > > >
> > > > **A5:** WAST, like most feature selection methods (Table 2 from [1]), provides the ranking of the input features. The user can select the desired number of features (K) based on the application and the target task's memory/computational resource limitation. Determining the optimal K is an open and challenging problem [1]. From the algorithmic side of feature selection, our proposed approach is independent of the chosen K. Section 4.4 and Figure 2 provide an analysis of the effect of algorithmic dependence on K in increasing the computational costs, as in the CAE method.
> > > >
> > > > [1] Li, Jundong, Kewei Cheng, Suhang Wang, Fred Morstatter, Robert P. Trevino, Jiliang Tang, and Huan Liu. "Feature selection: A data perspective." ACM computing surveys (CSUR) 50, no. 6 (2017): 1-45.

---

> > > > > ### Comment · Reviewer_U347 · 2022-08-08
> > > > > **Thank you for your extensive rebuttal**
> > > > >
> > > > > Most of my concerns were addressed by the authors' response. I raised the review score.

---

### Author Response · Authors · 2022-08-02
**Response to All Reviewers**

We thank all the reviewers for the positive reviews and constructive comments that help us to emphasize the contributions of our approach. We are encouraged to hear that the reviewers found the approach **novel** (Reviewers y8xW, XbZ9), the **gains are excellent/significant** (Reviewers cbLE, XbZ9), and the method is **superior across various domains** (Reviewers U347, cbLE). We appreciate that the reviewers acknowledge that the paper is well-presented. In response to the thoughtful comments, we have addressed them one by one in the individual responses and provided a summary below:

* We added analysis on the effect of the sparsity level of WAST on performance.
* We added analysis on the robustness of the studied methods to noisy samples.
* We added comparisons to another recently supervised NN-based method and unsupervised classical method.
* We performed additional experiments on a popular music dataset.

Thanks again for all of your valuable suggestions. We updated the paper accordingly, with changes highlighted in green. We appreciate the reviewers' time to check our response.

---

### Comment · Area_Chair_tAtF · 2022-08-08
**Reviewers U347, cbLE, y8xW, XbZ9  - please reply to the authors**

The authors have responded to questions that you raised:
- U347 requested a comparison to DUFS, pointed out that no experiments are done with noisy samples, claims that no tests are done on datasets having fewer samples than features, requested testing of other NN-based approaches, suggested that the number of selected features be a hyperparameter, requested 10 replications of each experiment instead of 5, requested a more detailed comparison to QS, flagged a typo in Table 2, asked if the other methods were reimplemented by the paper authors, requested numeric results on consistency and stability, asked about missing results in Table 2, suggested comparing training times for NN and non-NN methods, and asked about accuracy/sparsity tradeoffs
- cbLE asked about applicability of the method beyond classification, why only input and output neurons were subject to the drop-and-grow process, and why WAST improved classification accuracy
- y8xW asked how λ is selected, if the output is processed through a softmax, for a comparison to DropConnect, and suggested testing on something fresher than the spoken digit recognition task
- XbZ9 claims that WAST does not appear to add noise, expresses concern that WAST is competitive with supervised methods, and asks for an evaluation (on synthetic data) of whether or not WAST selects the correct features

As we are near the end of the author-reviewer discussion period, I ask that you please read the authors' reply and respond promptly.

Thanks

---

### Meta-Review · Area_Chair_tAtF · 2022-08-24

**Recommendation:** Accept
**Confidence:** Certain

**Metareview:**

After the rebuttal and discussion, all reviewers recommend acceptance of this paper to some degree. The paper has benefited from a careful review by reviewer U347 and additional experiments and clarifications performed by the authors in response to the reviewer's concerns. All reviewers noted that the paper is clearly written, the proposed method is simple and easily implemented, and that it appears to perform quite well.


**Award:**

No

---

### Decision · Program_Chairs · 2022-09-14

Accept